# Dynamo-Depth: Fixing Unsupervised Depth Estimation for Dynamical Scenes

**Yihong Sun**
Cornell University
yihong@cs.cornell.edu

**Bharath Hariharan**
Cornell University
bharathh@cs.cornell.edu

## Abstract

Unsupervised monocular depth estimation techniques have demonstrated encouraging results but typically assume that the scene is static. These techniques suffer when trained on dynamical scenes, where apparent object motion can equally be explained by hypothesizing the object's independent motion, or by altering its depth. This ambiguity causes depth estimators to predict erroneous depth for moving objects. To resolve this issue, we introduce Dynamo-Depth, an unifying approach that disambiguates dynamical motion by jointly learning monocular depth, 3D independent flow field, and motion segmentation from unlabeled monocular videos. Specifically, we offer our key insight that a good initial estimation of motion segmentation is sufficient for jointly learning depth and independent motion despite the fundamental underlying ambiguity. Our proposed method achieves state-of-the-art performance on monocular depth estimation on Waymo Open [34] and nuScenes [3] Dataset with significant improvement in the depth of moving objects. Code and additional results are available at https://dynamo-depth.github.io.

## 1 Introduction

Embodied agents acting in the real world need to perceive both the 3D scene around them as well as how objects around them might behave. For instance, a self-driving car executing a lane change will need to know where the nearby cars are and how fast they are moving. Instead of relying on expensive sensors such as LiDAR to estimate this structure, a promising alternative is to use commodity cameras. This has motivated a long line of work on monocular depth estimation using neural networks.

While neural networks can learn to estimate depth, the dominant training approach requires expensive 3D sensors in the training phase. This limits the amount of training data we can capture, resulting in downstream generalization challenges. One would prefer to train these depth estimators *without supervision*, for e.g., using unlabeled videos captured from a driving car. This is possible to do if the videos are produced by a camera moving in a static scene, since the apparent motion (i.e., optical flow) of each pixel is then inversely proportional to the depth of the pixel. This provides a useful supervisory signal for learning depth estimation and has been used effectively in the past [48, 36].

However, for unsupervised approaches to learn depth, *dynamic scenes* with moving objects pose a challenge. Here, the apparent pixels motion on the image plane is a combined effect of *camera ego-motion* (or rigid scene motion) and the *independent motion* of objects in the scene. Methods that ignore the presence of moving objects and treat the scene as static will then learn erroneous depth, where the depths of moving objects are altered to match their observed motion, as shown in Figure 1 (b). Past works have tried to overcome this issue by having another module predict independent object motion [31, 24, 15], but the presence of multiple equivalent solutions makes training difficult: even with sophisticated regularizers, the training can converge to the degenerate solution of either predicting the whole scene as static with incorrect depth, or predicting the entire scene as moving on a flat canvas. This results in depth error on moving objects at least 4 times as large as the error on static

37th Conference on Neural Information Processing Systems (NeurIPS 2023).

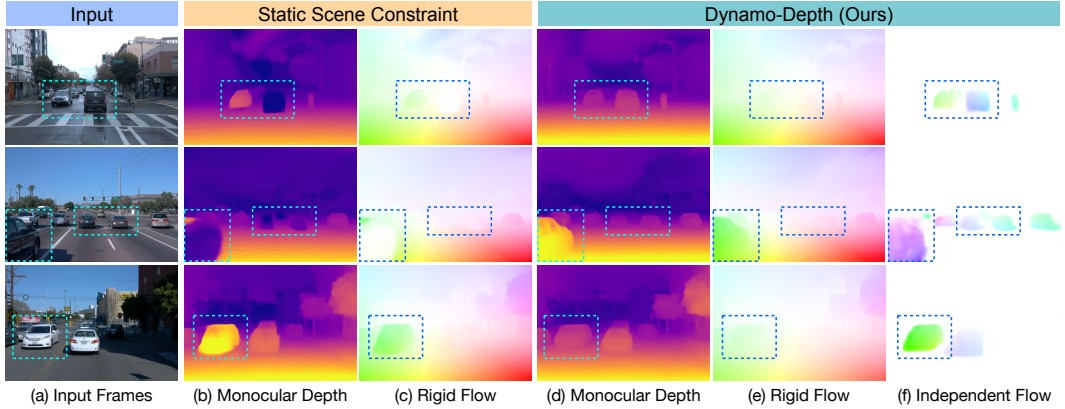

| Input | Static Scene Constraint | | Dynamo-Depth (Ours) | | |

(a) Input Frames     (b) Monocular Depth     (c) Rigid Flow     (d) Monocular Depth     (e) Rigid Flow     (f) Independent Flow

Figure 1: Visualization of erroneous cases of infinite object depth and floating objects that arise from the static scene constraint. The motion of the black SUVs and white van are explained via (b) *incorrect* depth predictions to generate (c) the correct rigid flow for reconstruction under the static scene constraint. In contrast, Dynamo-Depth predicts (d) *correct* monocular depth by explicitly disentangling (f) the 3D independent flow field from (e) the camera ego-motion induced rigid flow.

background even for state-of-the-art methods. Since navigating safely around moving objects is a primary challenge in self-driving, poor depth estimation of moving objects is particularly problematic.

In this paper, we propose a new architecture and training formulation for this unsupervised learning problem that substantially improves depth accuracy for moving objects. Our key insight is that we need a good initial estimate for which pixels are independently moving (a "motion mask"). With this motion mask at hand, one can easily train depth estimation using the static pixels, and account for independent motion to correctly estimate the depth of the moving objects. However, the challenge is obtaining this motion mask in the first place: how do we identify which pixels are independently moving? First, we introduce two separate network modules that predict *complete 3D scene flow* and *camera ego-motion*. The difference between these can then be used to identify the moving objects. Second, we introduce a novel initialization scheme that stops depth training early to prevent the depth estimator from learning erroneous depth for the moving objects. This allows the motion estimation networks to learn the correct scene flow. In sum, we make the following contributions:

1. We propose a new approach, Dynamo-Depth, for learning depth estimation for dynamic scenes solely from unlabeled videos. In addition to monocular depth, Dynamo-Depth also predicts camera ego-motion, 3D independent flow, and motion segmentation.

2. We propose a new architecture that identifies independent object motion via the difference between the rigid flow field induced by camera ego-motion and a complete scene flow field to facilitate learning and regularization.

3. We propose a novel motion initialization technique that learns an early estimation of motion segmentation to explicitly resolve the ambiguity and disentangle independent motion from camera ego-motion (Figure 1). This allows the entire architecture to learn end-to-end without any additional supervision from auxiliary model predictions or ground-truth labels.

4. We evaluate on both nuScenes [3] and Waymo Open [34] and demonstrate not only state-of-the-art performance in terms of overall depth accuracy and error, but also show large improvements on moving objects by up to 62% relative gain in accuracy and 68% relative reduction in error, while achieving up to 71.8% in F1 score in motion segmentation without any supervision.

## 2 Related Work

### 2.1 Unsupervised Monocular Depth Estimation

The framework for jointly learning monocular depth and ego-motion from unlabeled consecutive frames was first explored in [48, 36], where the objective was posed as a novel-view synthesis problem. Here, the monocular depth estimated from the target frame is coupled with the predicted ego-motion

to warp the reference frame to be photometrically consistent with the target frame. Following these works, 3D geometric consistency [28] and depth prediction consistency [1] were imposed across consecutive frames to further constrain the unsupervised learning objective. Furthermore, discrete volumes [19], uncertainty estimation [30], optical flow based pose estimation [47], minimal projection loss with auto-masking [10], and multiple input frames during inference [40, 25] were also proposed to improve depth predictions and mitigate influence of occlusion and relative stationary pixels.

**Unsupervised Scene Flow.** In addition to predicting depth from a single frame, a closely related task of scene flow estimation - obtaining both the 3D structure and 3D motion from two temporally consecutive images - can also be learned by leveraging the same novel-view reconstruction objective. When trained on sequences of stereo-frames, scene flow can be estimated from both stereo [18, 41] and monocular [16, 17] inputs during inference. Additionally, DRAFT [13] learned from monocular sequences only by utilizing synthetic data and multiple consecutive frames during inference.

## 2.2 Mitigating Static Scene Constraint

In essence, the reconstruction objective encourages pixels to be predicted at an appropriate distance away to explain its apparent motion on the image plane, when coupled with the predicted ego-motion. This assumes that all pixel motion can be sufficiently modeled by depth and camera ego-motion alone, which implies an underlying static scene. However, since dynamical objects are common in the wild, such static scene assumption is often violated. To disambiguate independent motion, recent works were proposed to leverage stereo-view information [38, 26, 11], additional input modalities [45, 8], synthetic data [13], supervised segmentation task [21], and auxiliary monocular depth network [33]. Extending from these methods, additional works proposed to mitigate the negative effects by modeling the behavior of dynamical objects without the use of additional modalities and labels.

**Modeling Independent Motion via Optical Flow.** Although an explainability mask [48] can be used to ignore the independently moving regions, recent works exploited the jointly learned optical flow to refine the depth prediction under non-rigid scene settings. This includes predicting a residual optical flow [43, 35] to account for dynamical objects and enforcing forward-backward flow consistency [49] to discount dynamical regions. EPC++ [27] directly modeled the dynamical objects by integrating its depth, ego-motion, and optical flow predictions. Additionally, CC [31] proposed a coordinated learning framework for joint depth and optical flow estimation assisted by a segmentation branch.

**Modeling Independent Motion via 3D Non-Rigid Flow Field.** Instead of modeling dynamical objects via 2D optical flow, additional works proposed to explicitly model their 3D independent flow field to encourage accurate depth learning for dynamical objects. To facilitate independent motion learning, various methods [4, 12, 22, 23, 2, 32] leveraged semantic priors from off-the-shelf pretrained detection or segmentation models to estimate regions that are "possibly moving." Additionally, Dyna-DepthFormer [46] used multi-frame information to compute the depth map, while iteratively refining the residual 3D flow field via a jointly trained motion network. Finally, Li *et al.* [24] proposed to jointly learn monocular depth, ego-motion, and residual flow field via a sparsity loss and RM-Depth [15] improved upon it via an outlier-aware regularization loss. Unlike [24, 15] where the residual/independent flow is predicted directly, we propose to model the 3D independent flow field via the complete scene flow, which facilitates training and effectively enforces sparsity.

# 3 Motivation and background

When a camera moves in a rigid scene, the apparent motion of image pixels is entirely explained by (a) their 3D position relative to the camera, and (b) the motion of the camera. This fact is used by unsupervised techniques that learn to predict depth from unlabeled videos captured as a camera moves through the scene (as in driving footage). Concretely, one network predicts depth from a target frame and another predicts the motion of the camera from the source frame to the target frame. The two predictions are then used to reconstruct a target frame from the source frame, and a reconstruction loss would serve as the primary training objective.

What happens if there are moving objects in the scene? Clearly, depth and camera-motion are not sufficient to explain the apparent image motion of these objects, resulting in a high reconstruction loss for these dynamical regions. In principle, the motion of these objects are unpredictable from a single frame, and so even though the network sees a high loss for these objects, the best it can do is to

fit the average case of a static object. As such, in principle, moving objects should not be a concern for learning-based techniques. However, in practical driving scenes, there is a large class of moving objects for which the network can in fact minimize the reconstruction loss by predicting an incorrect depth estimate. We describe this class below.

**Epipolar ambiguity in driving scenes:**

Consider the simple case of a pinhole camera moving forward along the Z axis (i.e., along its viewing direction). This is in fact the common case of a car driving down the road. In this case, the image location $(x, y)$ and corresponding optical flow $(\dot{x}, \dot{y})$ for a 3D point $(X, Y, Z)$ in the world would be:

$$x = \frac{X}{Z}, \qquad y = \frac{Y}{Z}, \qquad \dot{x} = \frac{-X\dot{Z}}{Z^2} = -\frac{\dot{Z}}{Z}x, \qquad \dot{y} = \frac{-Y\dot{Z}}{Z^2} = -\frac{\dot{Z}}{Z}y \qquad (1)$$

Consider what happens when this 3D point is on an object moving *collinear* to the camera (e.g., cars in front of the ego-car, or oncoming traffic). In this case, there are two contributions to $\dot{Z}$: the camera motion and the independent motion of the object. Thus, given ego-motion, one can produce the same optical flow by adding the appropriate amount of independent motion to $\dot{Z}$, or changing the depth $Z$. For example, an object can appear to move faster if it is either closer and static (i.e., $Z$ is smaller) or it is farther and moving towards the camera (i.e., $\dot{Z}$ is larger). This is a special case of *epipolar ambiguity* [42]. This implies that it is possible to model independent motion under the static scene constraint using an erroneous depth estimation. Unfortunately, there are statistical regularities that enable such erroneous learning.

**Statistical regularities:**

In self-driving scenes, there is often enough information in object appearance to predict its motion even from a single frame. The high capacity monocular depth network can learn to recover this information, and then alter its depth prediction to better reconstruct the target frame. For instance, in Figure 1 (b) the depth network may have learnt to recognize the backs of cars on the same or nearby lanes. In the training data, cars seen from the back often travel in the same direction with similar speed as the ego car, and thus have a very small optical flow in consecutive frames. The depth network therefore predicts that the car should be very far away; thus correctly predicting its optical flow, but incorrectly predicting its depth. A similar effect happens when the white van is recognized in its frontal view on the opposite lane, and the depth network predicts it to be closer to the ego-car. This allows the depth network to model the statistically-likely higher optical flow as a result of the object moving towards the camera. Furthermore, the speed of the object can also be roughly estimated from object identity (pedestrian < cyclist < car) and background context (residential < city < highway). Therefore, by recognizing the statistical regularities that give away object motion, the depth model can model the behavior of dynamical objects by predicting incorrect depth values to minimize the reconstruction objective.

This suggests that simply training a depth network on dynamic scenes is not a good solution, and an alternative framework is needed. However, we do note that learning these statistical regularities about object motion is much harder than learning the depth of the static background, since the background is a lot more consistent in its optical flow across videos and frames. Thus, we observe that this kind of overfitting to dynamical objects only happens later in training; a fact we use below.

## 4 Method

Following past works, we formulate the learning objective as a novel-view synthesis where the target frame is reconstructed from the source frame. As shown in Section 3, modeling the rigid flow induced by camera motion alone is not sufficient for modeling the behavior of dynamical objects and causes performance degradation for moving objects. We address this by learning a separate 3D flow field that captures the independent motion of dynamical objects.

### 4.1 Architecture

The proposed architecture (Figure 2) contains two components that collaborate together to explain the apparent pixels motion on the image plane. Section 4.1.1 details the rigid motion networks that model

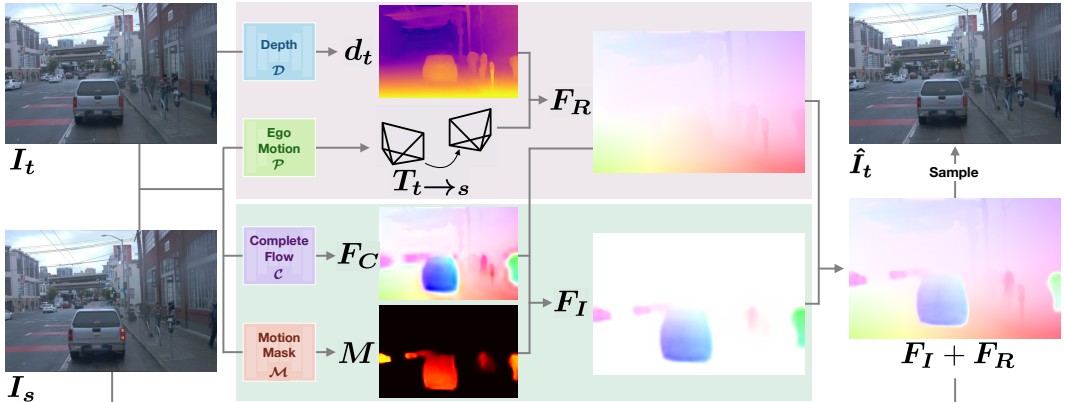

Figure 2: **Overview of the proposed Dynamo-Depth.** The target frame $I_t$ is reconstructed from source frame $I_s$ by composing the 3D rigid flow $F_R$ (predicted via Depth $\mathcal{D}$ & Pose $\mathcal{P}$) and 3D independent flow $F_I$ (predicted via Complete Flow $\mathcal{C}$ & Motion Mask $\mathcal{M}$). [1]

the 3D rigid flow field (motion induced by camera while assuming a static scene), which includes the depth network $\mathcal{D}$ and ego-motion pose network $\mathcal{P}$. Section 4.1.2 details the independent motion networks that model the 3D independent flow field (motion of dynamical objects), which includes the complete flow network $\mathcal{C}$ and motion mask network $\mathcal{M}$.

### 4.1.1 Rigid Motion Networks: Depth $\mathcal{D}$ & Pose $\mathcal{P}$

Given a source frame $I_s$ and a target frame $I_t$, we compute the 3D rigid flow induced by camera motion, $F_R(p_t)$, for each target pixel $p_t$ as follows. We first obtain the monocular depth map $d_t$ by passing $I_t$ to the depth network $\mathcal{D}$ (Eq. 2). We pass $I_s$ and $I_t$ to the ego-motion pose network $\mathcal{P}$ to obtain the relative camera pose $T_{t \rightarrow s}$ (rotation and translation) (Eq. 3). We then use the depth and inverse camera intrinsics $K^{-1}$ to back-project every target pixel $p_t$ to compute its 3D location $P_t$, and transform $P_t$ into its corresponding 3D location, $\hat{P}_s$, in the source frame via $T_{t \rightarrow s}$. (Eq. 4). The "rigid flow" induced by camera motion is then simply $F_R(p_t) = \hat{P}_s - P_t$. [2]

$$\mathcal{D} : \mathbb{R}^{H \times W \times 3} \rightarrow \mathbb{R}^{H \times W} \qquad\qquad d_t = \mathcal{D}(I_t) \qquad\qquad (2)$$

$$\mathcal{P} : \mathbb{R}^{H \times W \times 3} \times \mathbb{R}^{H \times W \times 3} \rightarrow \mathbb{R}^{3 \times 4} \qquad T_{t \rightarrow s} = \mathcal{P}(I_t, I_s) \qquad\qquad (3)$$

$$P_t = d_t(p_t) K^{-1} \overrightarrow{p_t}, \quad \hat{P}_s = T_{t \rightarrow s} \overrightarrow{P_t}, \qquad F_R(p_t) = \hat{P}_s - P_t \in \mathbb{R}^3 \qquad (4)$$

### 4.1.2 Independent Motion Networks: Complete Flow $\mathcal{C}$ & Motion Mask $\mathcal{M}$

In addition to the rigid flow $F_R$, we also wish to model object motion via the 3D independent flow, denoted as $F_I$. With independent motion explicitly modeled by $F_I$, we reformulate $\hat{P}_s$ as $\hat{P}_s = (F_I(p_t) + F_R(p_t) + P_t)$ to integrate the contribution of the independent flow vector $F_I(p_t)$. However, directly estimating $F_I$ via a network turns out to be difficult for two reasons:

**(a) Learning.** During training, the network will need to fill in the missing independent motion vector that is appropriate for the current separately predicted monocular depth and ego-motion for a correct reconstruction: a "moving target" during training. Furthermore, it is intuitively much more difficult to learn and predict independent motion directly when the apparent motion in the input frames consists of both rigid and independent motion entangled together, especially in the ambiguous scenarios illustrated in Section 3.

**(b) Regularization.** Adding to this complexity, $F_I(p_t)$ should be non-zero if and only if the pixel $I_t(p_t)$ is independently moving. One could regularize this network to encourage sparsity, but early on when depth and ego-motion predictions are noisy, a high sparsity term would encourage this

---

[1] $F_C$, $F_R$, $F_I$ and $F_I + F_R$ are all 3D flow fields visualized as optical flow on the image plane.
[2] $\overrightarrow{\cdot}$ denotes the conversion from Cartesian to homogeneous coordinate.

network to turn off entirely. At the other end of the spectrum, a weak sparsity regularization allows independent flow $F_I$ to be active on static regions of the scene, which would "explain away" the rigid motion $F_R(p_t)$ when computing $\hat{P}_s$ and corrupt the learning signal for depth estimation. Also, conventional sparsity regularization on $F_I$ (e.g., $L_{1/2}$ sparsity loss [24]) would dampen the magnitude of predicted motion, which reduces $F_I$'s capacity to model fast or far-away moving objects.

To address these issues, we propose to decompose the prediction of independent flow $F_I(p_t)$ in two: (1) predicting whether pixel $I_t(p_t)$ is independently moving and (2) predicting how it is moving. For (1), we model the probability of independent motion as a binary motion segmentation task where motion mask network $\mathcal{M}$ takes the consecutive frames $I_t$ and $I_s$ and predict a per-pixel motion mask $M$ as shown in Eq. 5. For (2), instead of directly predicting the independent motion, we predict the complete scene flow $F_C(p_t)$ via a complete flow network $\mathcal{C}$ as shown in Eq. 6.

$$\mathcal{M} : \mathbb{R}^{H \times W \times 3} \times \mathbb{R}^{H \times W \times 3} \to \mathbb{R}^{H \times W} \qquad M = \sigma(\mathcal{M}(I_t, I_s)) \qquad (5)$$

$$\mathcal{C} : \mathbb{R}^{H \times W \times 3} \times \mathbb{R}^{H \times W \times 3} \to \mathbb{R}^{H \times W \times 3} \qquad F_C = \mathcal{C}(I_t, I_s) \qquad (6)$$

From these predictions, we formulate the independent flow field as $F_I = M \cdot (F_C - F_R)$, i.e., the residual flow between $F_C$ and $F_R$ gated by the motion mask $M$. Intuitively, computing $\hat{P}_s = (F_I(p_t) + F_R(p_t) + P_t)$ can be thought of as combining the contribution of the complete flow $F_C(p_t)$ and rigid flow $F_R(p_t)$ as a weighted average via $M(p_t)$ (Eq. 8).

$$F_I = M \cdot (F_C - F_R) \qquad (7)$$

$$\hat{P}_s = F_I(p_t) + F_R(p_t) + P_t = M(p_t) \cdot F_C(p_t) + (1 - M(p_t)) \cdot F_R(p_t) + P_t \qquad (8)$$

This formulation directly resolves the two aforementioned issues. For (a), $\mathcal{C}$ has a simpler learning objective of modeling the full scene flow with per-pixel predictions, i.e. $\hat{P}_s = F_C(p_t) + P_t$. For (b), the sparsity regularization can be applied to $M$ alone without any impact to the learning of $\mathcal{C}$ and its ability to learn the complete scene flow.

Finally, we compute the reconstruction $\hat{I}_t(p_t)$ by sampling $I_s$ at the corresponding pixel coordinate $\hat{p}_s$ via $\vec{\hat{p}_s} \equiv K\hat{P}_s$ for every target pixel $p_t$.

## 4.2 Loss Function

We present the overall loss function in Eq. 9. The photometric loss $L_{recon}$ in Eq. 10 serves as the main learning objective, which evaluates the reconstruction $\hat{I}_t$ via SSIM [39] and L1 weighted by $\alpha$.

$$L = L_{recon} + L_s + \gamma_c L_c + \gamma_m L_m + \gamma_g L_g \qquad (9)$$

$$L_{recon} = \frac{\alpha}{2}(1 - \text{SSIM}(I_t, \hat{I}_t)) + (1 - \alpha)||I_t - \hat{I}_t||_1 \qquad (10)$$

The smoothness loss $L_s$ in Eq. 11 regularizes the smoothness of the predicted depth map $d_t$, complete flow $F_C$, and motion mask $M$. The edge-aware smoothness loss $l_s(z, I) = |\delta_x z| \exp(-|\delta_x I|) + |\delta_y z|| \exp(-|\delta_y I|)$ is weaker around pixels with high color variation. We also use the mean-normalized inverse depth $d_t^*$ to discourage shrinking of the estimated depth [37].

$$L_s = \gamma_{sd} l_s(d_t^*, I_t) + \gamma_{sc} l_s(F_C, I_t) + \gamma_{sm} l_s(M, I_t) \qquad (11)$$

The motion consistency loss $L_c$ in Eq. 12 computes the flow discrepancy $F_D(p)$ between the complete scene flow $F_C(p)$ and rigid flow $F_R(p)$ for static pixels. The probability of pixel $p$ being static is approximated by $(1 - M(p))$.

$$L_c = \frac{1}{HW} \sum_p (1 - M(p)) \cdot F_D(p), \quad F_D(p) = ||F_C(p) - F_R(p)||_1 \qquad (12)$$

The motion sparsity loss $L_m$ in Eq. 13 considers all pixels with flow discrepancy $F_D$ lower than the mean value over the frame as putative background, and encourages the motion mask to be 0 by using the cross entropy against label $\vec{0}$, which is denoted as $g(\cdot)$.

$$L_m = g\big(\{M(p) : \forall p \text{ s.t. } F_D(p) \leq \text{mean}(F_D)\}\big) \qquad (13)$$

Finally, the above-ground loss $L_g$ in Eq. 14 penalizes projected 3D points below the estimated ground plane (via RANSAC) where $d_t^g$ denotes the mean-normalized inverse depth of the ground plane.

$$L_g = \frac{1}{HW} \sum_p \text{ReLU}(d_t^g(p) - d_t^*(p)) \qquad (14)$$

### 4.3 Motion Initialization

Since the reconstruction loss is the main learning signal in the unsupervised approach, the gradient from the reconstruction loss between $I_t$ and $\hat{I}_t$ at pixel $p_t$ would propagate through the computed sample coordinate $\overrightarrow{p_s} \equiv K\hat{P}_s \equiv K(F_I(p_t) + F_R(p_t) + P_t)$. Here, although it may be ideal to jointly train all models end-to-end from start to finish, in practice, updating the rigid flow vector $F_R(p_t)$ and the independent flow vector $F_I(p_t)$ jointly from the start would cause the learning problem to be ill-posed, since each component has the capacity to overfit and explain away the other's prediction. Due to the ambiguity between camera motion and object motion as discussed in Section 3, the image reconstruction error alone is not a strong enough supervisory signal to enforce the accurate prediction of all underlying sub-tasks jointly, as a *family* of solutions of depth and object motion would all satisfy the training objective, ranging from predicting the whole scene as static with incorrect depth to predicting the scene as moving objects on a flat canvas.

Here, we offer our key insight that a good initial estimate of motion segmentation $M$ would allow the system to properly converge without arising degenerate solutions. As shown in Eq. 8, $\hat{P}_s = M(p_t) \cdot F_C(p_t) + (1 - M(p_t)) \cdot F_R(p_t) + P_t$. For any pixel $p_t$, $\hat{P}_s \approx F_C(p_t) + P_t$ if $M(p_t)$ is near 1, and approximates $F_R(p_t) + P_t$ if $M(p_t)$ is near 0. From this, it is clear that a good motion mask would route the back-propagating gradient to the complete flow network $\mathcal{C}$ via $F_C$ if $p_t$ is moving and route the gradient to the rigid motion networks $\mathcal{D}$ and $\mathcal{P}$ via $F_R(p_t)$ if $p_t$ is static.

Of course, the question is how one can initialize a good motion mask, since it requires identifying the moving objects. Ideally, moving objects would be identified as the pixels that are poorly reconstructed based on depth and ego-motion alone, but as discussed in Section 3, the depth network has the capacity to alter its depth predictions and reconstruct even the moving objects. Nevertheless, we observe that overfitting to the moving objects in this way is difficult, and only happens in later iterations. In earlier training iterations, reconstruction errors for dynamical pixels are ignored (Appendix B.5).

To ensure a good motion mask $M$, we first train a depth network assuming a static scene, but stop updates to the depth network at an early stage. We then initialize the independent motion networks $\mathcal{C}$ and $\mathcal{M}$ from a coarse depth sketch predicted by this frozen early-stage depth network. Thus, without access to any labels or auxiliary pretrained networks, we obtain an initialization for motion mask $M$ that helps in disambiguating and accurately estimating the rigid flow $F_R$ and independent flow $F_I$.

## 5 Experiments

**Dataset.** While Dynamo-Depth is proposed for general applications where camera motion is informative for scene geometry, we focus our attention on the challenging self-driving setting, where the epipolar ambiguity and the statistical regularities mentioned in Section 3 that lead to erroneous depth predictions are highly prevalent. Specifically, we evaluate on three datasets - Waymo Open [34], nuScenes [3], and KITTI [9] with Eigen split [6].

On all datasets, we only use the unlabeled video frames for training; ground truth LiDAR depth is only used for evaluation. We evaluate both overall depth estimation accuracy, as well as accuracy computed separately for static scene and moving objects. For the latter evaluation, we use Waymo Open and nuScenes and identify moving objects by rectifying the panoptic labels with 3D box labels to obtain masks for static/moving objects. It is worth quantifying the number of moveable objects per frame in each dataset. Waymo Open has a mean of $12.12$ with a median of $9$. In nuScenes, the mean is $7.78$ with a median of $7$. In KITTI, due to lack of per-frame labels, we approximate using its curated object detection dataset, which only has a mean of $5.26$ with a median of $4$. Additional dataset information and distribution histograms are found in Appendix C.

**Model and Training setup.** The proposed method is trained on four NVIDIA 2080 Ti with a total batch size of 12 and an epoch size of 8000 sampled batches. Adam optimizer [20] is used with an initial learning rate of 5e-5 and drops to 2.5e-5 after 10 epochs. Motion Initialization lasts 5 epochs and takes place after the depth network and complete flow network have been trained for 1 epoch each. After initialization, the system is trained for 20 epochs, totalling approximately 20 hours. The hyperparameter values are the same for all experiments and are provided in Appendix B.1, along with the rest of the details of the model architecture. To demonstrate the generality of the proposed framework, we adopt two architectures for the depth network $\mathcal{D}$, one with a ResNet18 [14] backbone

Table 1: Depth evaluation on the KITTI (K), nuScenes (N), and Waymo Open (W) Dataset. *IM* stands for independent motion where ✗ denotes a lack of independent motion modeling. *Sem* indicates the amount of semantic information given during training, where 'm' indicates mask-level supervision and 'b' indicates box-level supervision. Manual replication with released code is indicated by [†].

| | IM | Sem | D | Error metric (↓) | | | | Accuracy metric (↑) | | |
| | | | | Abs Rel | Sq Rel | RMSE | RMSE log | $\delta < 1.25$ | $\delta < 1.25^2$ | $\delta < 1.25^3$ |
|---|---|---|---|---|---|---|---|---|---|---|
| Monodepth2 [10] | ✗ | | K | 0.115 | 0.882 | 4.701 | 0.190 | 0.879 | 0.961 | 0.982 |
| LiteMono [44] | ✗ | | K | **0.101** | 0.729 | **4.454** | **0.178** | **0.897** | **0.965** | 0.983 |
| Struct2Depth [4] | | m | K | 0.141 | 1.026 | 5.290 | 0.215 | 0.816 | 0.945 | 0.979 |
| Dyna-DM [32] | | m | K | 0.115 | 0.785 | 4.698 | 0.192 | 0.871 | 0.959 | 0.982 |
| Lee *et al.* [23] | | b | K | 0.114 | 0.876 | 4.715 | 0.191 | 0.872 | 0.955 | 0.981 |
| SGDepth [21] | | m | K | 0.113 | 0.835 | 4.693 | 0.191 | 0.879 | 0.961 | 0.981 |
| Boulahbal *et al.* [2] | | m | K | 0.110 | 0.719 | 4.486 | 0.184 | 0.878 | 0.964 | **0.984** |
| GeoNet [43] | | | K | 0.155 | 1.296 | 5.857 | 0.233 | 0.793 | 0.931 | 0.973 |
| EPC++ [27] | | | K | 0.141 | 1.029 | 5.350 | 0.216 | 0.816 | 0.941 | 0.976 |
| CC [31] | | | K | 0.140 | 1.070 | 5.326 | 0.217 | 0.826 | 0.941 | 0.975 |
| TrianFlow [47] | | | K | 0.113 | 0.704 | 4.581 | 0.184 | 0.871 | 0.961 | **0.984** |
| Li *et al.* [24] | | | K | 0.130 | 0.950 | 5.138 | 0.209 | 0.843 | 0.948 | 0.978 |
| RM-Depth [15] | | | K | 0.107 | **0.687** | 4.476 | 0.181 | 0.883 | 0.964 | **0.984** |
| **Dynamo-Depth** (MD2) | | | K | 0.120 | 0.864 | 4.850 | 0.195 | 0.858 | 0.956 | 0.982 |
| **Dynamo-Depth** | | | K | 0.112 | 0.758 | 4.505 | 0.183 | 0.873 | 0.959 | **0.984** |
| Monodepth2[†] [10] | ✗ | | N | 0.425 | 16.592 | 10.040 | 0.402 | 0.723 | 0.837 | 0.887 |
| LiteMono[†] [44] | ✗ | | N | 0.419 | 15.578 | 9.807 | 0.449 | 0.720 | 0.831 | 0.879 |
| **Dynamo-Depth** (MD2) | | | N | 0.193 | 2.285 | 7.357 | 0.287 | 0.765 | 0.885 | 0.935 |
| **Dynamo-Depth** | | | N | **0.179** | **2.118** | **7.050** | **0.271** | **0.787** | **0.896** | **0.940** |
| Monodepth2[†] [10] | ✗ | | W | 0.173 | 2.731 | 7.708 | 0.227 | 0.797 | 0.930 | 0.968 |
| LiteMono[†] [44] | ✗ | | W | 0.158 | 2.305 | 7.394 | 0.210 | 0.816 | 0.944 | 0.976 |
| Struct2Depth [4] | | m | W | 0.180 | 1.782 | 8.583 | 0.244 | - | - | - |
| Li *et al.* [24] | | m | W | 0.157 | 1.531 | 7.090 | 0.205 | - | - | - |
| Lee *et al.* [23] | | b | W | 0.148 | 1.686 | 7.420 | 0.210 | - | - | - |
| Li *et al.* [24] | | | W | 0.162 | 1.711 | 7.833 | 0.223 | - | - | - |
| **Dynamo-Depth** (MD2) | | | W | 0.130 | 1.439 | 6.646 | 0.183 | 0.851 | 0.959 | 0.985 |
| **Dynamo-Depth** | | | W | **0.116** | **1.156** | **6.000** | **0.166** | **0.878** | **0.969** | **0.989** |

from *Monodepth2* [10] and another with a CNN/Transformer hybrid backbone from *LiteMono* [44], denoted as *Dynamo-Depth (MD2)* and *Dynamo-Depth*, respectively. To be commensurate with baselines, the encoders are initialized with ImageNet [5] pretrained weights.

**Metrics.** Depth performance is reported via commonly used metrics proposed in [7], including 4 error metrics (Abs Rel, Sq Rel, RMSE, and RMSE log) and 3 accuracy metrics ($\delta < 1.25$, $\delta < 1.25^2$, and $\delta < 1.25^3$). We also report precision-recall curve for evaluating binary motion segmentation.

## 5.1 Monocular Depth Estimation

As shown in Table 1, our proposed approach outperforms prior arts on both nuScenes and Waymo Open across all metrics, with over 57% and 21% reduction in overall Abs Rel for nuScenes and Waymo Open, respectively. This improvement is significant: we repeat the experiment for *Dynamo-Depth* on Waymo Open 3 times and obtained a 95% confidence interval of $0.119 \pm 0.003$ for Abs Rel and $0.874 \pm 0.004$ for $\delta < 1.25$. This suggests the challenge and importance of estimating accurate depth for moving objects in realistic scenes. On KITTI, our approach is competitive with prior art. However, KITTI has fewer moveable objects, which better conforms to the static scene constraint and diminishes the importance in modeling independent motion. For instance, *LiteMono* [44], which does not model independent motion, demonstrates superior performance on KITTI but drastically underperforms in nuScenes and Waymo Open, with both datasets having many more moving objects.

To further evaluate the effectiveness of our approach in modeling independent motion, we split nuScenes and Waymo Open into static background, static objects and moving objects and evaluate depth estimation performance on each partition. As we adopt our depth model from *Monodepth2* [10] and *LiteMono* [44], in Table 2, we compare our approach against the respective baseline with the same depth architecture. For simplicity, we report Abs Rel and $\delta < 1.25$. Notably, by explicitly modeling

Table 2: Depth evaluation with semantic split on the nuScenes (N) and Waymo Open (W) Dataset. *S.B.*, *S.O.* and *M.O.* denotes the partition of pixels that are static background, static moveable object and moving object, respectively. Manual replication with released code is indicated by [†].

| | D | Abs Rel (↓) | | | | $\delta < 1.25$ (↑) | | | |
|---|---|---|---|---|---|---|---|---|---|
| | | *All* | *S.B.* | *S.O.* | *M.O.* | *All* | *S.B.* | *S.O.* | *M.O.* |
| Monodepth2[†] [10] | N | 0.425 | 0.447 | 0.232 | 0.418 | 0.723 | 0.735 | 0.704 | 0.570 |
| Dynamo-Depth (MD2) | N | 0.193 | 0.196 | 0.196 | 0.228 | 0.765 | 0.761 | 0.759 | 0.684 |
| **Δ (%)** | | **-54.6** | **-56.2** | **-15.5** | **-45.5** | **+5.8** | **+3.5** | **+7.8** | **+20.0** |
| LiteMono[†] [44] | N | 0.419 | 0.431 | 0.248 | 0.502 | 0.720 | 0.734 | 0.718 | 0.517 |
| Dynamo-Depth | N | 0.179 | 0.184 | 0.182 | 0.198 | 0.787 | 0.781 | 0.774 | 0.753 |
| **Δ (%)** | | **-57.3** | **-57.3** | **-26.6** | **-60.6** | **+9.3** | **+6.4** | **+7.8** | **+45.6** |
| Monodepth2[†] [10] | W | 0.173 | 0.152 | 0.215 | 0.749 | 0.797 | 0.810 | 0.683 | 0.416 |
| Dynamo-Depth (MD2) | W | 0.130 | 0.122 | 0.175 | 0.234 | 0.851 | 0.862 | 0.778 | 0.674 |
| **Δ (%)** | | **-24.9** | **-19.7** | **-18.6** | **-68.8** | **+6.8** | **+6.4** | **+13.9** | **+62.0** |
| LiteMono[†] [44] | W | 0.158 | 0.140 | 0.179 | 0.599 | 0.816 | 0.827 | 0.733 | 0.506 |
| Dynamo-Depth | W | 0.116 | 0.110 | 0.155 | 0.194 | 0.878 | 0.891 | 0.812 | 0.750 |
| **Δ (%)** | | **-26.6** | **-21.4** | **-13.4** | **-67.6** | **+7.6** | **+7.7** | **+10.8** | **+48.2** |

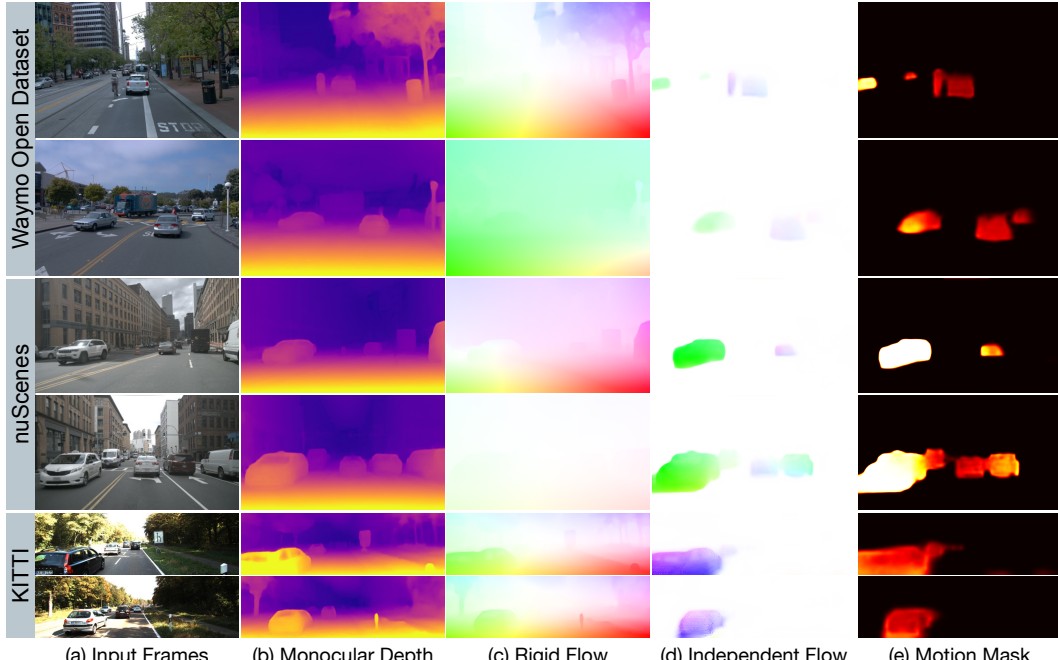

(a) Input Frames    (b) Monocular Depth    (c) Rigid Flow    (d) Independent Flow    (e) Motion Mask

Figure 3: Qualitative results of our proposed approach that learns from (a) unlabeled video sequences and predicts (b) monocular depth, (c) rigid flow, (d) independent flow, and (e) motion segmentation. [3]

independent motion, we observe a consistent and significant improvement on moving objects (*M.O.*) across both architectures and both datasets, with over $48\%$ relative improvement in accuracy and over $67\%$ relative reduction in error for both architectures on Waymo Open. Additionally, we see a substantial improvement on both static moveable objects (*S.O.*) and static background (*S.B.*) for both datasets and both architectures. Interestingly, the large reduction in error on static background in nuScenes does not match the corresponding improvement in accuracy, as the large errors on background mainly come from night-time instances where all methods fail, but differ in their respective artifacts (discussed further in Appendix D.2).

---

[3]All motion segmentation in (e) are uniformly visualized with a range of [0, 1], and corresponding video sequences that contain each example are found in `https://dynamo-depth.github.io`.

Table 3: Depth ablation results on the Waymo Open Dataset with Dynamo-Depth.

| $L_c$ | Motion Initialization | $L_g$ | ImageNet Pretraining | Abs Rel (↓) | | | | $\delta < 1.25$ (↑) | | | |
|---|---|---|---|---|---|---|---|---|---|---|---|
| | | | | *All* | *S.B.* | *S.O.* | *M.O.* | *All* | *S.B.* | *S.O.* | *M.O.* |
| ✗ | | | | 0.525 | 0.489 | 0.625 | 0.447 | 0.254 | 0.274 | 0.210 | 0.325 |
| | ✗ | | | 0.522 | 0.487 | 0.621 | 0.444 | 0.256 | 0.275 | 0.212 | 0.326 |
| | | ✗ | | 0.266 | 0.237 | 0.167 | 0.211 | 0.816 | 0.844 | 0.792 | 0.743 |
| | | | ✗[2] | 0.136 | 0.132 | 0.177 | 0.201 | 0.833 | 0.839 | 0.778 | 0.730 |
| | | | | 0.116 | 0.110 | 0.155 | 0.194 | 0.878 | 0.891 | 0.812 | 0.750 |

In sum, by explicitly modeling independent motion, we are able to outperform the respective baselines and achieve state-of-the-art performances on both Waymo Open and nuScenes Dataset. Qualitative results are found in Figure 1 and Figure 3, where "Static Scene Constraint" refers to performance of *LiteMono* [44] in Figure 1.

## 5.2 Motion Segmentation and Method Ablation

Figure 4 evaluates the quality of the jointly learned binary motion mask network using precision and recall. We observe a high precision with increasing recall, achieving a F1 score of $71.8\%$ on Waymo Open [34]. Note that this segmentation quality is achieved without any labels. On nuScenes, Dynamo-Depth achieves a F1 score of $42.8\%$ with $57.2\%$ on day-clear conditions and $21.1\%$ on all other conditions. Notably, the motion mask network is able to segment out small dynamical objects, such as the cyclist in the first row of Figure 3.

In addition, we show our ablation results on Waymo Open in Table 3. First, we found that ablating the motion consistency loss removes the regularization on the complete flow prediction, which causes the degenerate solution of moving objects on a flat canvas, as indicated by the poor segmentation performance in Figure 4. Similarly, by ablating motion initialization, initial independent motion prediction lead to large reconstruction errors, which diverges depth training. In addition, we observe that ablating the above ground prior preserves the performance of motion segmentation and depth of moving objects, but degrades the performance of depth on static background. Intuitively, this prior regularizes the structural integrity of the ground plane. Finally, we observed that the original set of hyperparameter values leads to training divergence when ablating the ImageNet pretrained weights. To compensate, we increase the initial number of epochs for depth learning from 1 to 2 (marked as ✗[2]) without performing any hyperparameter sweeps.

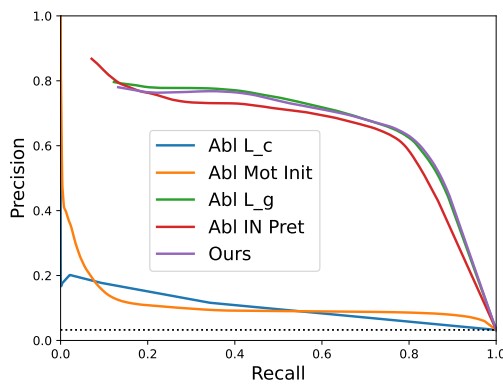

Figure 4: Precision-Recall curve for motion segmentation on the Waymo Open Dataset.

## 6 Conclusion

In this work, we identify the effects of dynamical objects in unsupervised monocular depth estimation with static scene constraint. To mitigate the negative impacts, we propose to jointly learn depth, ego-motion, 3D independent motion and motion segmentation from unlabeled videos. Our key insight is that a good initial estimation of motion segmentation encourages joint depth and independent motion learning and prevents degenerate solutions from arising. As a result, our approach is able to achieve state-of-the-art performance on both Waymo Open and nuScenes Dataset.

**Limitations and Societal Impact.** Our work makes a brightness consistency assumption, where the brightness of a pixel will remain the same. Although common in optical flow, this assumption limits the our method's ability to model scenes with dynamical shadows, multiple moving light sources, and lighting phenomenons (e.g., lens flare). In addition, our work does not introduce any foreseeable societal impacts, but will generally promote more label-efficient and robust computer vision models.

# 7 Acknowledgement

This research is based upon work supported in part by the National Science Foundation (IIS-2144117 and IIS-2107161). Yihong Sun is supported in part by an NSF graduate research fellowship.

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

# A Visualizations

In Figure 5, we show qualitative comparison between *LiteMono* [44] and our Dynamo-Depth. We select absolute relative difference (*Abs Rel* in quantitative results) as the measure of error, as shown in Eq. 15 with predicted depth $\boldsymbol{d}$, ground truth depth $d^*$, and pixel coordinate $p$.

$$\text{Abs Rel}(p) = \begin{cases} |\boldsymbol{d}(p) - d^*(p)|/d^*(p), & \text{if } p \text{ is a LiDAR point} \\ 0, & \text{otherwise} \end{cases} \tag{15}$$

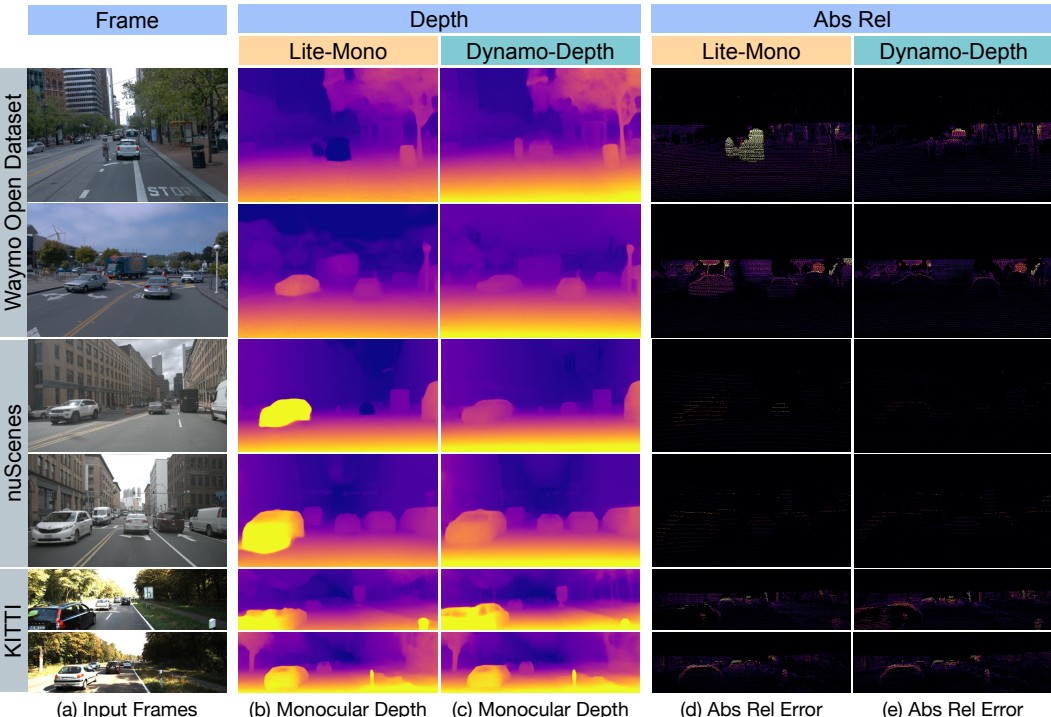

Figure 5: Qualitative comparison between *LiteMono* [44] and our proposed Dynamo-Depth. The input frame is shown in column (a). Column (b) and (c) show the depth predictions and column (d) and (e) show the absolute relative difference for each LiDAR point with a fixed error range of $[0, 1]$.

# B Implementational Details

## B.1 Hyperparameters

Please refer to Table 4 for the hyperparameters used for training. The values are found via Waymo Open and kept the same for other datasets. The trade-off coefficient $\alpha$ and depth smoothness coefficient $\gamma_{sd}$ are set to $0.85$ and $0.001$, respectively, as in [10]. The motion smoothness coefficient $\gamma_{sc}$ is set to $0.001$ to be consistent with $\gamma_{sd}$ and was not tuned further. The mask smoothness coefficient $\gamma_{sm}$ is set to a much higher value of $0.1$ to remove sporadic false positives induced by high frequency image features such as trees and buildings. Since both rigid motion $\boldsymbol{F_R}$ and complete motion $\boldsymbol{F_C}$ have relatively small magnitudes, we set the motion consistency coefficient $\gamma_c$ to be $5.0$ to properly enforce consistency. Due to the high loss values outputted by binary cross entropy, we set the mask sparsity coefficient to be $0.04$ to accommodate. The above ground coefficient $\gamma_g$ is set to $0.1$ after searching across the values $0.01$, $0.1$, and $1.0$.

## B.2 Architectural Details

The depth network $\mathcal{D}$ is adopted from two architectures, one with a ResNet18 [14] backbone from *Monodepth2* [10] and another with a CNN/Transformer hybrid backbone from *LiteMono* [44], denoted

Table 4: List of hyperparameters used.

| Hyperparameter | Value |
|---|---|
| $\alpha$ | 0.85 |
| $\gamma_{sd}$ | 0.001 |
| $\gamma_{sc}$ | 0.001 |
| $\gamma_{sm}$ | 0.1 |
| $\gamma_c$ | 5.0 |
| $\gamma_m$ | 0.04 |
| $\gamma_g$ | 0.1 |

as *Dynamo-Depth (MD2)* and *Dynamo-Depth*, respectively. For both versions of our model, the pose network $\mathcal{P}$ is adopted from Monodepth2 [10]. The encoders of the complete flow network $\mathcal{C}$ and motion mask network $\mathcal{M}$ are shared and have the same architecture as $\mathcal{P}$. To obtain per-pixel predictions, the features of the shared encoder are fed into the the separate decoders with the same architecture as the depth decoder in Monodepth2 [10] and with output per-pixel dimension of 3 and 1 for $\mathcal{C}$ and $\mathcal{M}$, respectively.

### B.3 Training Details

For notational simplicity, Figure 2 only considers the case of a single source frame $I_s$, while in practice, Dynamo-Depth is trained with two source frames: one frame previous and one frame after the target $I_t$. Here, the two respective reconstructions are rectified via a minimal projection loss proposed in [10] to obtain $\hat{I}_t$. To enforce forward-backward consistency, the independent motion networks $\mathcal{C}$ and $\mathcal{M}$ predict a single independent flow field $F_I$ that is used to warp both source frames into the target frame, with the appropriate sign change.

Following *Monodepth2* [10] and *LiteMono* [44], we perform auto-masking to facilitate learning during depth initialization. Afterwards, auto-masking is turned off as the apparently static pixels may belong to dynamical objects traveling with the ego-camera, in which case auto-masking would mask out the corresponding reconstruction loss essential for independent motion learning.

There is a linear weight ramp for all loss terms that include $F_C$ and $M$ at the beginning of each training stage, with the length of 2666 iterations for Waymo and nuScenes, and 8000 iterations for KITTI due to the its limited number of moving objects.

### B.4 Ground Plane Estimation

Concretely, 3D projected points ($P_t$ in Eq. 4) belonging to the bottom half of the image are sampled to construct the ground plane $d_t^g$ via RANSAC (5 points are selected per iteration, for a total of 100 iterations). This assumption does break under cases of traffic jams, but due to the rare occurrence of these scenarios for this learning task assuming a moving camera, the negative effects are limited.

### B.5 Difficulties in Overfitting to Dynamical Objects

In Figure 6, we show a qualitative example where a depth network learned with static scene constraint (*LiteMono* [44] in this case) would ignore the high reconstruction loss in earlier iterations in (b) and overfit to the moving objects via erroneous depth predictions only in later iterations in (c)-(f).

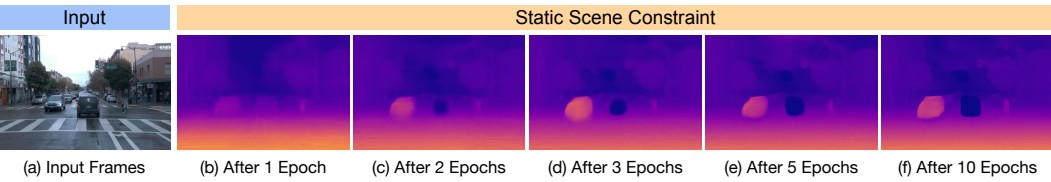

Figure 6: Visualization of the overfitting phenomenon as shown via *LiteMono* [44], where erroneous depth for the black SUV and the oncoming car is only learned in later iterations, in (c)-(f).

## C   Dataset Details

We provide additional information of the three datasets used for evaluation.

For the Waymo Open Dataset [34], 76,852 front camera image-triplets from the provided train set containing 798 video sequences are used for training while 2,216 front camera images uniformly sampled from the provided validation set containing 202 video sequences are used for evaluation. During training, frames are downsampled to $480 \times 320$. We rectify the panoptic labels [29] with 3D box labels to obtain the corresponding masks for static and moving objects.

For the nuScenes Dataset [3], 79,760 front camera image-triplets from the provided train set containing 700 video sequences are used for training while 6019 front camera images from the official validation set containing 150 video sequences are used for evaluation (of which 4449 are *day-clear*). Following [8], we consider a sequence to be in the *day-clear* subset if its description does not contain "night" or "rain". During training, frames are downsampled to $512 \times 288$. We rectify the panoptic LiDAR labels with annotated 3D bounding boxes to obtain motion attributes for LiDAR points.

Notably, as our method learns to explicitly model independent motion, there is no filtering based on camera velocity recordings when constructing the training dataset.

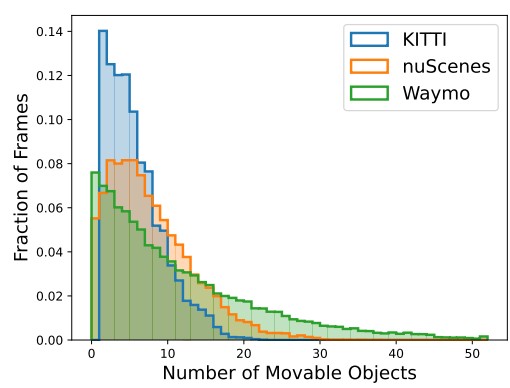

Figure 7: Histogram of the number of moveable objects per frame for all three dataset.

For the KITTI Dataset [9], we follow the Eigen split [6] with 39,180 image-triplets used for training and 697 images used for evaluation. During training, frames are downsampled to $640 \times 192$.

Finally, we plot the histogram of the number of moveable objects per frame for all three datasets. Shown in Figure 7, Waymo Open Dataset has the most moveable objects with a mean of $12.12$, followed by nuScenes with a mean of $7.78$. Due to the lack of per-frame object labels in KITTI, we evaluate on its curated object detection dataset, which has a mean of $5.26$. We note that the real distribution should be much tighter near $0$, compared to the detection dataset with all images containing at least $1$ moveable object.

Table 5: Additional depth evaluation on the KITTI Dataset. *Stereo Train* and *Stereo Infer.* stand for the use of stereo-view during training and inference, respectively. *Multi-Frame* denotes the use of multiple frames as inputs during inference. *Add. Sup.* stands for additional supervision during training where I, S, and D stand for IMU-level, synthetic-data, and pretrained depth model supervision, respectively.

| | Stereo | | Multi- | Add. | Error metric (↓) | | | | Accuracy metric (↑) | | |
|---|---|---|---|---|---|---|---|---|---|---|---|
| | Train | Infer. | Frame | Sup. | Abs Rel | Sq Rel | RMSE | RMSE log | $\delta < 1.25$ | $\delta < 1.25^2$ | $\delta < 1.25^3$ |
| UnOS [38] | ✓ | ✓ | | | 0.049 | 0.515 | 3.404 | 0.121 | 0.965 | 0.984 | 0.992 |
| EffiScene [18] | ✓ | ✓ | | | 0.049 | 0.522 | 3.461 | 0.120 | 0.961 | 0.984 | 0.992 |
| Xiang *et al.* [41] | ✓ | ✓ | | | 0.048 | 0.487 | 3.447 | 0.117 | 0.964 | 0.985 | 0.992 |
| Liu *et al.* [26] (stereo) | ✓ | ✓ | | | 0.051 | 0.532 | 3.780 | 0.126 | 0.957 | 0.982 | 0.991 |
| Liu *et al.* [26] (mono) | ✓ | | | | 0.108 | 1.020 | 5.528 | 0.195 | 0.863 | 0.948 | 0.980 |
| Hur *et al.* [16] | ✓ | | | | 0.125 | 0.978 | 4.877 | 0.208 | 0.851 | 0.950 | 0.978 |
| PLADE-Net [11] | ✓ | | | | 0.089 | 0.590 | 4.008 | 0.172 | 0.900 | 0.967 | 0.985 |
| DRAFT [13] | | | ✓ | S | 0.097 | 0.647 | 3.991 | 0.169 | 0.899 | 0.968 | 0.984 |
| Dyna-DepthFormer [46] | | | ✓ | | 0.094 | 0.734 | 4.442 | 0.169 | 0.893 | 0.967 | 0.983 |
| ManyDepth [40] | | | ✓ | | 0.090 | 0.713 | 4.261 | 0.170 | 0.914 | 0.966 | 0.983 |
| DepthFormer [25] | | | ✓ | | 0.090 | 0.661 | 4.149 | 0.175 | 0.905 | 0.967 | 0.984 |
| Zhang *et al.* [45] | | | | I | 0.108 | 0.761 | 4.608 | 0.187 | 0.883 | 0.962 | 0.982 |
| SC-DepthV3 [33] | | | | D | 0.118 | 0.756 | 4.709 | 0.188 | 0.864 | 0.960 | 0.984 |
| Dynamo-Depth | | | | | 0.112 | 0.758 | 4.505 | 0.183 | 0.873 | 0.959 | 0.984 |

Table 6: Depth evaluation on the nuScenes Dataset with *day-clear* split. *IM* stands for independent motion where ✗ denotes a lack of independent motion modeling. *Sup.* indicates additional supervision given during training, where 'V' indicates camera velocity recordings and 'R' indicates radar scans. Manual replication with released code is indicated by †.

| | *IM* | *Sup.* | Error metric (↓) | | | | Accuracy metric (↑) | | |
| --- | --- | --- | --- | --- | --- | --- | --- | --- | --- |
| | | | Abs Rel | Sq Rel | RMSE | RMSE log | $\delta < 1.25$ | $\delta < 1.25^2$ | $\delta < 1.25^3$ |
| Monodepth2† [10] | ✗ | | 0.157 | 2.432 | 6.943 | 0.248 | 0.844 | 0.940 | 0.968 |
| LiteMono† [44] | ✗ | | 0.161 | 2.618 | 6.818 | 0.251 | 0.846 | 0.939 | 0.966 |
| R4Dyn-L [8] | | V,R | 0.130 | 1.658 | 6.536 | - | 0.858 | - | - |
| **Dynamo-Depth** (MD2) | | | 0.149 | 1.792 | 6.536 | 0.236 | 0.830 | 0.938 | 0.972 |
| **Dynamo-Depth** | | | 0.132 | 1.571 | 6.158 | 0.216 | 0.856 | 0.951 | 0.977 |

Table 7: Depth evaluation with semantic split on the nuScenes Dataset with *day-clear* split. *S.B.*, *S.O.* and *M.O.* denotes the partition of pixels that are static background, static moveable object and moving object, respectively. Manual replication with released code is indicated by †.

| | Abs Rel (↓) | | | | $\delta < 1.25$ (↑) | | | |
| --- | --- | --- | --- | --- | --- | --- | --- | --- |
| | *All* | *S.B.* | *S.O.* | *M.O.* | *All* | *S.B.* | *S.O.* | *M.O.* |
| Monodepth2† [10] | 0.157 | 0.150 | 0.214 | 0.324 | 0.844 | 0.850 | 0.774 | 0.626 |
| Dynamo-Depth (MD2) | 0.149 | 0.146 | 0.186 | 0.197 | 0.830 | 0.834 | 0.779 | 0.737 |
| **Δ (%)** | **-5.1** | **-2.7** | **-13.1** | **-39.2** | **-1.7** | **-1.9** | **+0.6** | **+17.7** |
| LiteMono† [44] | 0.161 | 0.151 | 0.232 | 0.373 | 0.846 | 0.853 | 0.781 | 0.572 |
| Dynamo-Depth | 0.132 | 0.130 | 0.168 | 0.167 | 0.856 | 0.857 | 0.795 | 0.804 |
| **Δ (%)** | **-18.0** | **-13.9** | **-27.6** | **-55.2** | **+1.2** | **+0.5** | **+1.8** | **+40.6** |

# D    Additional Results

## D.1    Additional KITTI Comparisons

Please refer to Table 5 for additional comparisons on KITTI, where the compared methods leverage either additional modalities or additional supervision beyond the use of semantic information as shown in Table 1.

## D.2    Additional nuScenes Comparisons

Please refer to Table 6 and Table 7 for depth performance comparisons on the nuScenes validation *day-clear* subset, along with precision-recall curve for motion segmentation in Figure 8. Note that R4Dyn-L [8] is only trained on the *day-clear* subset with velocity filtering, while the rest of the methods are all trained with the entire dataset without any filtering. Furthermore, Table 7 still demonstrates the trend as observed in Table 2, where the a significant improvement on moving objects are observed in terms of both accuracy and error. Finally, as shown in Figure 9, both methods fail under low-light conditions where the brightness consistency assumption no longer holds. Due to the rigid scene constraint, artifacts appear in the respective depth prediction in (b), which causes a much larger error compared to the prediction in (d), as discussed in Section 5.1.

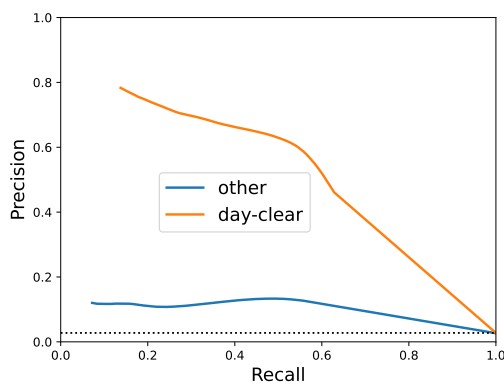

Figure 8: Precision-Recall curve for motion segmentation on the nuScenes Dataset, which is partitioned into the *day-clear* and *other* subsets.

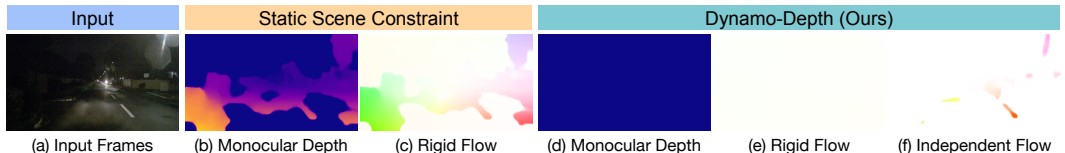

Figure 9: Visualizing predictions under night-time conditions in the nuScenes dataset.

## D.3 Odometry Evaluation

Please refer to Table 8 for odometry evaluation for Dynamo-Depth and Dynamo-Depth (MD2) on both nuScenes *day-clear* subset and Waymo Open Dataset.

Table 8: Odometry results on the nuScenes (*day-clear* subset) and Waymo Open Dataset. Results show the average absolute trajectory error over all overlapping 5-frame snippets and 100-frame snippets in the test sequences, and standard deviation, in meters. For consistency, the first 100 test sequences for both datasets are evaluated. Manual replication with released code is indicated by [†].

| Dataset | nuScenes | | Waymo | |
| --- | --- | --- | --- | --- |
| Speed (m/s) | $5.657 \pm 4.188$ | | $6.978 \pm 6.028$ | |
| Snippet Length (# frames) | 5 | 100 | 5 | 100 |
| Snippet Duration (seconds) | 0.3 | 7.4 | 0.4 | 9.9 |
| Snippet Counts | 22820 | 13320 | 19364 | 9864 |
| Monodepth2[†] [10] | $0.016 \pm 0.017$ | $0.097 \pm 0.069$ | $0.018 \pm 0.029$ | $0.143 \pm 0.194$ |
| Dynamo-Depth (MD2) | $0.019 \pm 0.029$ | $0.112 \pm 0.160$ | $0.015 \pm 0.027$ | $0.091 \pm 0.121$ |
| LiteMono[†] [44] | $0.018 \pm 0.018$ | $0.090 \pm 0.065$ | $0.022 \pm 0.044$ | $0.156 \pm 0.194$ |
| Dynamo-Depth | $0.017 \pm 0.019$ | $0.081 \pm 0.070$ | $0.017 \pm 0.036$ | $0.107 \pm 0.164$ |

## D.4 Scene Flow Comparisons

We were not able to compare against the mentioned related works in unsupervised scene flow estimation due to the large difference in methodology (stereo-view during training [18, 41, 16, 17]) and lack of codebase for reproducibility [13].

## E Limitations

As shown in Figure 10, dynamical shadows and moving light sources in low-light conditions violate the brightness consistency assumption, which results in erroneous predictions of motion segmentation and monocular depth, respectively.

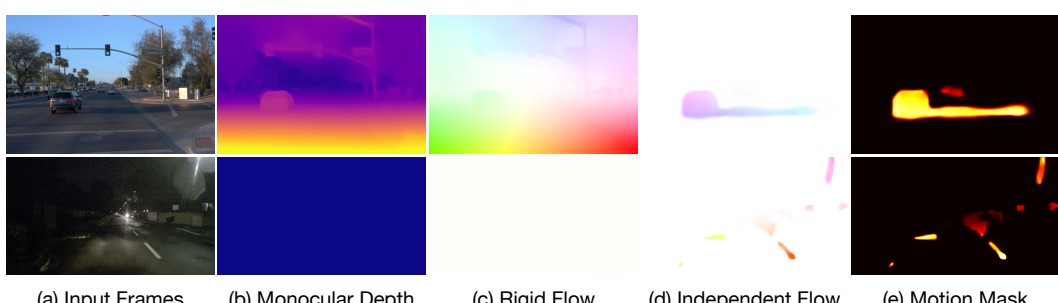

Figure 10: Limitations of Dynamo-Depth.

