# OpenReview forum: "Dynamo-Depth: Fixing Unsupervised Depth Estimation for Dynamical Scenes"
_NeurIPS.cc/2023/Conference — NeurIPS 2023 poster_

### Official Review · Reviewer_9pRa · 2023-07-05

**Soundness:** 4 excellent
**Presentation:** 3 good
**Contribution:** 3 good
**Rating:** 7
**Confidence:** 4

**Summary:**

This paper introduces a new unsupervised approach and a new network architecture for monocular depth estimation.

The paper is motivated by the necessity of new algorithms capable of learning to predict depth from unlabelled data while being able to perform reliable predictions in the regions of moving objects, which is particularly critical for autonomous driving applications.

To tackle this problem, the authors proposed a new architecture that separates the observable flow field as rigid regions and independently moving regions. A training scheme is also proposed to effectively learn this disentanglement guided by a motion masking that is also learned in a self-supervised manner.

The proposed approach is evaluated on three datasets and achieves state of the art results on nuScenes and Waymo Open, specifically improving by 50% on moving objects.

**Strengths:**

The main insight in the proposed method is to decompose the flow field used to obtain depth supervision as rigid flow F_R, which goal is to explain the flow field due to the camera motion as if the scene where rigid, and independent flow F_I, which models the flow field resulting from moving objects. Decomposing these two factors is not trivial and the paper proposes an interesting solution to the problem that does not require annotated labels.

The insights presented about the later overfitting of depth estimation models to dynamically moving objects is a very interesting finding and supports the main idea in the paper. The authors successfully explored this observation to propose a new training scheme for learning the motion mask without requiring segmentation labels.

The results are convincing both quantitatively and qualitatively, as provided in the supplementary material, specially for moving objects. The paper is well written and relatively easy to follow.


**Weaknesses:**

As in many previous works, the proposed approach targets the depth estimation problem as a novel-view synthesis problem. The particular case of moving objects is then tackled by a separated 3D flow field that captures the independent motion of moving objects, which is coherently discussed in the paper. Although this setup seems to be generic, the proposed approach is only evaluated in the context of autonomous driving. Therefore, it is not evidenced in the paper if the proposed formulation would adapt well to different contexts if a moving camera. For instance, if the camera is moving in a way that results in no parallax effect, could this framework diverge to a solution with most of the pixels being segmented as moving objects?


The main contribution of the paper is in the network architecture and specially in the training strategy. However, the idea of braking the scene rigidity assumption by estimating a motion field for moving objects was already explored in [11] and the notion of sparse moving objects in the scene as a prior was also explored in [17]. The main differences with these two works in special are only briefly mentioned in the paper.



**Questions:**

How the ground plane d^g_t used in Eq. 13 is defined? The authors mentioned that it is estimated via RANSAC, but is this assuming that the ground is mostly empty and visible? What happens in traffic jam situations? Please clarify this part of the method and provide what are the assumptions for the ground plane estimation.

Typos:
L48 "learning erroneous depth for the moving objects and allow."
L51 " achieve achieve"
L198: F_D instead of F_d ?

**Limitations:**

Limitations were briefly discussed in the paper and no major concerns are remaining.

---

> ### Author Rebuttal · Authors · 2023-08-10
>
> Thank you for your support and constructive comments! We are encouraged that you find the paper well-written, the insights interesting, and the results convincing! We list our response to your questions below.
>
> ---
> ### Contexts Beyond Self-Driving
> Our approach would work well in other general settings where camera motion is informative for scene geometry. Even when there are some videos without parallax effects, if **the majority of the dataset contains meaningful camera motion**, then the model would recover from the ambiguity in these edge cases and work as intended.
>
> If the camera motion results in **no parallax effect at all** for the entire dataset, all scenes would become ambiguous and the geometry becomes impossible to recover. To clarify, the camera motion that does not produce parallax effect would correspond to rotation about the pin-hole. In this case, for our proposed approach, due to the sparsity regularization term (Eq. 13), it is more likely for the model to predict every point in the scene to be static without any independent motion (assuming the apparent motion is ego-motion-induced only).
>
> Regarding our choice of self-driving data, to the best of our knowledge, we could not find any large real dataset that captures dynamical scenes in the wild for training. Therefore, we focus our attention on the challenging self-driving setting, where the epipolar ambiguity and the statistical regularities that lead to erroneous depth predictions are highly prevalent.
>
> - Epipolar Ambiguity: Collinearly moving objects are common, since all agents in nearby lanes are moving in the same or opposite direction as the camera.
>
> - Statistical Regularities: From one frame, an object’s appearance can indicate its direction of motion (frontal vs. rear view) and a rough estimate of speed (car vs. cyclist).
>
> We will incorporate discussions regarding different contexts and a more general setting in the final version.
>
> ---
> ### Comparison against Li et. al [17] and RM-Depth [11]
> Our proposed method differs from both Li et. al [17] and RM-Depth [11] in the following ways:
>
> - Our method learns a separate motion segmentation network, whereas Li et. al [17] and RM-Depth [11] keep it implicit. By explicitly predicting dynamical regions as opposed to implicitly thresholding the independent motion field in [11,17], our method has a better capacity for learning semantically meaningful motion segmentation, instead of modeling low-level pixel motion outliers.
>
> - Unlike [11,17] where the residual/independent motion field is predicted directly, we model the independent motion field $F_I$ by learning the complete scene flow $F_C$. Unlike the independent motion field $F_I$, which also requires accurate ego-motion for correct reconstruction, the complete scene flow $F_C$ is much easier to learn since it is independent of the predicted ego-motion, removing its dependence on the quality of the ego-motion prediction during training. Intuitively, it is much more difficult to learn and predict independent motion directly when the observed motion in the input consists of both rigid and non-rigid motion entangled together.
>
> Specifically compared to Li et. al [17]:
>
> - Li et. al [17] applies a $L_{1/2}$ sparsity loss on the independent motion field which directly penalizes large motion that is common in the real world, whereas we apply the sparsity loss on the motion segmentation network without suppressing the independent motion field.
>
> - As a result, we achieve superior performance on Waymo against Li et. al [17], reducing the Absolute Relative difference from 0.162 to 0.116.
>
> Specifically compared to RM-Depth [11]:
>
> - RM-depth only reports performance on KITTI and Cityscapes without releasing code for reproducibility so we cannot compare against its ability in modeling moving objects in more chaotic settings such as Waymo or nuScenes. (We reached out to the author but our request for a code base for only evaluation purposes was denied.)
>
> Thank you for pointing this out, as we will include additional discussion in the final draft.
>
> ---
> ### Ground Plane Estimation
> Concretely, 3D projected points ($P_t$ in Eq.5) belonging to the bottom half of the image are sampled to construct the ground plane $d^g_t$ via RANSAC (5 points are selected per iteration, for a total of 100 iterations). It is true that traffic jams break this assumption, but due to the rare occurrence of these scenarios for this learning task assuming a moving camera, the negative effects are limited. Thank you and we will clarify this point in the final version.
>
> ---
> ### Typos
> Thank you for pointing out these typos and we will correct them in the final version.

---

### Official Review · Reviewer_mnoy · 2023-07-06

**Soundness:** 1 poor
**Presentation:** 2 fair
**Contribution:** 1 poor
**Rating:** 3
**Confidence:** 5

**Summary:**

This paper aims to estimate the depth map in a dynamic scene environment.
To this end, the paper proposed 1) two separate module architectures that estimate rigid scene flow and residual scene flow and 2) a motion initialization method to enable a stable learning process.
However, the technical and performance comparisons with the previous self-supervised depth estimation, self-supervised scene flow estimation, self-supervised motion segmentation, and self-supervised optical flow are insufficient.
The current manuscript needs a lot of modification.


**Strengths:**

- The proposed method achieves better performance in nuScenes and Waymo datasets. However, the comparison group is too limited to judge the superiority of the proposed method.

**Weaknesses:**

W1. Technical and performance comparisons with the recent self-supervised scene flow estimation

The proposed method needs to describe its originality and superiority compared to recent scene flow estimation on the KITTI, nuScene, or Waymo datasets.

- Xiang, Xuezhi, et al. "Self-supervised learning of scene flow with occlusion handling through feature masking." Pattern Recognition 139 (2023): 109487.
- Jiao, Yang, Trac D. Tran, and Guangming Shi. "Effiscene: Efficient per-pixel rigidity inference for unsupervised joint learning of optical flow, depth, camera pose and motion segmentation." Proceedings of the IEEE/CVF Conference on Computer Vision and Pattern Recognition. 2021.
- Hur, Junhwa, and Stefan Roth. "Self-supervised monocular scene flow estimation." Proceedings of the IEEE/CVF Conference on Computer Vision and Pattern Recognition. 2020.
- Hur, Junhwa, and Stefan Roth. "Self-supervised multi-frame monocular scene flow." Proceedings of the IEEE/CVF Conference on Computer Vision and Pattern Recognition. 2021.
- Guizilini, Vitor, et al. "Learning optical flow, depth, and scene flow without real-world labels." IEEE Robotics and Automation Letters 7.2 (2022): 3491-3498.

W2. Technical and performance comparisons with the recent self-supervised motion segmentation

The proposed method needs to describe its originality and superiority compared to recent motion segmentation on the KITTI, nuScene, or Waymo datasets.

- Liu, Liang, et al. "Unsupervised Learning of Scene Flow Estimation Fusing with Local Rigidity." IJCAI. 2019.
- Jiao, Yang, Trac D. Tran, and Guangming Shi. "Effiscene: Efficient per-pixel rigidity inference for unsupervised joint learning of optical flow, depth, camera pose and motion segmentation." Proceedings of the IEEE/CVF Conference on Computer Vision and Pattern Recognition. 2021.
- Xiang, Xuezhi, et al. "Self-supervised learning of scene flow with occlusion handling through feature masking." Pattern Recognition 139 (2023): 109487.

W3. Technical and performance comparisons with the recent self-supervised optical flow estimation

The proposed method needs to describe its originality and superiority compared to recent optical flow estimation on the KITTI, nuScene, or Waymo datasets.

- Teed, Zachary, and Jia Deng. "Raft: Recurrent all-pairs field transforms for optical flow." Computer Vision–ECCV 2020: 16th European Conference, Glasgow, UK, August 23–28, 2020, Proceedings, Part II 16. Springer International Publishing, 2020.
- Jonschkowski, Rico, et al. "What matters in unsupervised optical flow." Computer Vision–ECCV 2020: 16th European Conference, Glasgow, UK, August 23–28, 2020, Proceedings, Part II 16. Springer International Publishing, 2020.
- Zhao, Wang, et al. "Towards better generalization: Joint depth-pose learning without posenet." Proceedings of the IEEE/CVF Conference on Computer Vision and Pattern Recognition. 2020.

W4. Technical and performance comparisons with the recent self-supervised depth estimation

The proposed method needs to describe its originality and superiority compared to recent self-supervised depth estimation on the KITTI, nuScene, or Waymo datasets.

- Watson, Jamie, et al. "The temporal opportunist: Self-supervised multi-frame monocular depth." Proceedings of the IEEE/CVF Conference on Computer Vision and Pattern Recognition. 2021.
- PLADE-Net: Towards Pixel-Level Accuracy for Self-Supervised Single-View Depth Estimation with Neural Positional Encoding and Distilled Matting Loss, Proceedings of the IEEE/CVF Conference on Computer Vision and Pattern Recognition. 2021.
- Guizilini, Vitor, et al. "Multi-frame self-supervised depth with transformers." Proceedings of the IEEE/CVF Conference on Computer Vision and Pattern Recognition. 2022.


**Questions:**

My major questions are shown in the weakness part.

**Limitations:**

This paper must include in-depth technical and performance comparisons with the recent self-supervised depth, scene flow, optical flow, and motion segmentation literature.
The current manuscript doesn't support the proposed method's originality or superiority without the comparisons.

---

> ### Author Rebuttal · Authors · 2023-08-10
>
> Thank you for taking the time to point out the references!  We outline the differences between our approach and each of the references below. We have added many of these references to Table 8 (see below for details) and we also discuss how we propose to add additional evaluations to our final camera-ready.
>
> ---
> ### Depth Estimation
>
> Among the three mentioned works, PLADE-Net uses **stereo-view during training**, while ManyDepth and DepthFormer uses **multiple frames during inference**. In comparison, our depth model is trained from monocular videos only and takes a single frame as input during inference. The following bullet points detail the differences in comparison to our proposed work.
>
> - PLADE-Net uses **stereo input** during training, which is a different task from our method which learns from monocular videos only.
>
> - ManyDepth (Watson et al.) is proposed as a cost volume based approach to fuse temporal information from **multiple frames as input**, whereas our method only requires a single frame. To alleviate the effect of moving objects, ManyDepth trains a single-frame inference-time monocular depth network and leverages it as a teacher network. However, the teacher network would be susceptible to the same issues as we detailed in Section 3, making it unreliable in more complex scenes (Figure 1).
>
> - DepthFormer (Guizilini et al.) uses **multiple frames as input**, whereas our method only requires a single frame.
>
> Nevertheless, for all the above mentioned papers that evaluate depth estimation performance, we have now added their results into Table 8 (Rebuttal PDF). We will include this new comparison and the above discussion in the final camera ready.
>
> ---
> ### Scene Flow Estimation
> All five mentioned works leverage either stereo information or additional synthetic data, while our model learns from **monocular** videos only **without any labels or semantic knowledge** from pretrained models and predicts complete scene flow along with monocular depth, independent motion field, and motion segmentation. Please refer to the following bullet points for additional discussions.
> - Xiang et al. uses **stereo information** during both training and inference, while our approach has no access to stereo information.
>
> - Similarly, EffiScene (Jiao et al.) uses **stereo information** during both training and inference, while our method predicts depth from monocular view only.
>
> - Hur et al. 2020 uses **stereo information during training**, while our method is strictly monocular.
>
> - Hur et al. 2021 builds on top of Hur et al. 2020 and also uses **stereo information during training**.
>
> - DRAFT (Guizilini et al.) **utilizes additional synthetic data with synthetic ground truth**, whereas our method only learns from the unlabeled monocular videos.
>
> In spite of the different problem setup, we include depth results from all of these papers (with the exception of Hur et al. 2021, which does not report depth) in Table 8. We will add these to the main paper.
>
> We will also include scene flow comparisons to these papers in the camera ready. However, we believe that this comparison must be done on Waymo and NuScenes because of the lack of moving objects in KITTI as pointed out by *Reviewer EvaZ*, L237-L239 in the manuscript, and Fig. 6 in the Rebuttal PDF. We will perform this comparison and add results to the main paper.
>
> ---
> ### Motion Segmentation
> All three mentioned works leverage **stereo information** during training, with EffiScene (Jiao et al.)  and Xiang et al. using **stereo information** during inference. This gives significant information to depth without needing to learn from camera motion during training. Once depth can be estimated, the independent motion that is entangled with rigid-motion, especially collinear with camera motion, can be disambiguated as well. In contrast, in our work, we are able to learn object motion and motion segmentation jointly with depth from monocular videos only without using any labels.
>
> We will incorporate additional discussions on this in the final version. In addition, we will also include quantitative comparisons where possible. Once again, we believe that this must be done on Waymo and NuScenes as discussed above.
>
> ---
> ### Optical Flow
>
> For optical flow, we would like to note that optical flow is not a focus of our work. We have no network that directly predicts optical flow, and we have no loss functions that encourage accurate optical flow to emerge (e.g. forward-backward consistency). Although we can obtain an indirect optical flow from our model by combining the outputs of our various modules, we do not claim that this will yield state-of-the-art on optical flow benchmarks.

---

> > ### Comment · Reviewer_mnoy · 2023-08-19
> >
> > Thank you for the author's detailed feedback. I carefully read the other reviews and all the rebuttals.
> > I keep my initial rating. The main reasons are as follows.
> >
> > 1) Insufficient evaluation: The main idea is motion decomposition. By explicitly modeling camera ego-motion and moving object motion, the networks learn more accurate geometry, and the self-supervision (e.g., image reconstruction loss) provides accurate supervision.
> > Therefore, the performance of motion segmentation is a key factor in improving the overall network's performance, including the monocular depth estimation network. Also, motion segmentation is highly related to the scene flow estimation network. However, none of the evaluations are reported in the current manuscript.
> > Without the experiments (at least, motion segmentation and scene flow), It is not sure whether the proposed method is truly effective or superior to the previous methods.
> >
> > 2) Novelty: The author claims the contributions are 1) the method can learn from monocular videos only without any labels or semantic knowledge from pretrained model, and 2) a motion initialization technique that resolves the ambiguity between camera motion and independent motion that frequently appears in complex dynamical scenes.
> >
> > However, the motivation for using unlabeled monocular images only without a pre-trained model is not convincing. Basically, the current method starts from the ImageNet pre-trained backbone. Also, recently, lots of foundation models have been provided. Adopting them and starting from them is not a big problem. Especially, synthetic data is easier to get than unlabeled images. Pre-trained model in the synthetic data will be a good starting point. Also, due to the limited comparison in Nuscenes and Waymo datasets, it is not sure whether the utilization of unlabeled monocular images only is superior or comparable with the previous methods that use pre-trained knowledge.
> >
> > Paradoxically, contribution 2 is originated from that the method doesn't use any pre-trained model.
> > The joint training of depth, scene flow, and ego-motion is an ill-posed problem.
> > Especially, training the networks from scratch makes the training process highly unstable and underperformed.
> > The known solutions in the previous joint training methods are 1) adopting pre-trained networks with the synthetic/labeled data and 2) utilizing multiple-stage training (e.g., flow->depth->joint depth-flow training).
> > However, it is not sure whether the current motion initialization is superior or comparable with the multiple-stage training method.
> >
> > In summary, the key idea of the paper is motion decomposition. However, the idea is not new. Also, the evaluations of motion segmentation and scene flow are missing. The proposed contributions are not rigidly supported by the experiments. For the nuscene and waymo datasets, more extensive comparisons, including recent methods, are essentially required. The current manuscript needs a lot of modification. Therefore, I keep my initial rating.

---

> > > ### Author Response · Authors · 2023-08-21
> > > **Response to mnoy**
> > >
> > > Thank you for your response. Since you raised a few new concerns, we would like to briefly address them here.
> > >
> > > ---
> > > ### Motion segmentation evaluation is essential and missing.
> > > The objective of our paper is improving unsupervised monocular depth performance for dynamical scenes, so we evaluate depth performance and demonstrate our efficacy in Table 1 and Table 2 of our manuscript.
> > >
> > > Additionally, we do show the precision recall curves for motion segmentation on Waymo Open as an ablation study in Figure 4 of our manuscript, where our method achieves **over 70% precision at 70% recall**. This demonstrates the effectiveness of our motion segmentation network that is jointly trained without any supervision.
> > >
> > > Finally, since all three mentioned works for motion segmentation use stereo information during training, we believe that stereo information would give substantial information to depth, which simplifies motion disentanglement.
> > >
> > > ---
> > > ### Lack of motivation for not using pretrained networks
> > > As a clarification, the ImageNet weights used as initialization are chosen only for a consistent comparison with prior works. It is fundamentally different from the use of frozen auxiliary pretrained networks that are trained for instance segmentation or depth estimation.
> > >
> > > In general, the use of pretrained networks would limit learning to be defined by the data distribution used to train the auxiliary network. Factors such as train-test domain gap and evolving domain shift across time would negatively impact learning. The auxiliary pretrained networks may fail to recognize vehicles that are unique to a certain region and under-represented in the training dataset. For these instances, a jointly learned motion segmentation from the in-distribution data would be superior to leveraging external pretrained networks.
> > >
> > > We disagree with the point that synthetic data is easier to obtain than unlabeled data. For tasks like depth estimation, synthetic data would need to include a diverse array of real world objects which is difficult to obtain, resulting in the same domain gap issues as with the use of pretrained networks.
> > >
> > > In general, multi-stage training would not directly disentangle independent motion from rigid motion, since the optical flow prediction would encompass both. Therefore, explicit disengagement of motion remains to be necessary in order to mitigate the influence of independent motion.
> > >
> > > ---
> > > ### Motion decomposition is not new
> > > Although unsupervised motion segmentation is not new, as far as we are aware, our method is the first that jointly learns motion segmentation with monocular depth, ego-motion, and independent motion solely from unlabeled videos.

---

> ### Comment · Reviewer_EvaZ · 2023-08-14
>
> Although adding the mentioned experiments could make the paper stronger, I do not think that comparing the proposed method with scene flow, optical flow, and motion segmentation is necessary. These results are just side outputs, and the final goal of this paper is depth estimation. The authors have conducted quite sufficient experiments on depth estimation, especially on Waymo and nuSecens datasets, which is large-scale and challenging. As far as I know, the evaluation is much more comprehensive than almost previous work.

---

### Official Review · Reviewer_EvaZ · 2023-07-06

**Soundness:** 4 excellent
**Presentation:** 4 excellent
**Contribution:** 4 excellent
**Rating:** 7
**Confidence:** 5

**Summary:**

This paper proposes a self-supervised depth estimation method. It addresses dynamic objects in the scene by jointly learning monocular depth, 3D independent flow field, and motion segmentation from unlabeled videos. The paper writing is very good, and the motivation/insights are clearly stated. The method is evaluated in KITTI, Waymo, and nuSecens datasets. Overall, I think that it is a high-quality paper.

**Strengths:**

The paper is well-written, and it is easy to read.

The results are evaluated in Waymo and nuSecens datasets. This is very good because the KITTI dataset has been overfitted, and it contains fewer moving objects than the other two datasets.

The Sec 3 is very informative. The discussion on motivations makes the paper easy to understand.

The evaluation results and ablation studies clearly demonstrate the efficacy of the proposed method.


**Weaknesses:**


There is no obvious weakness to me, but I think that it would be better if the authors could show visual odometry results. The nuSences and Waymo datasets provide camera poses, so it is not hard to do.

[a] also proposes a method for dynamic objects, but it uses a different solution. The comparison is not required, but it would be better if making a discussion on different ideas for addressing dynamic objects.

[a] SC-DepthV3: Robust Self-supervised Monocular Depth Estimation for Dynamic Scenes, arXiv 2022


**Questions:**

NA

---

> ### Author Rebuttal · Authors · 2023-08-10
>
> Thank you for your support and for finding our paper well-written and the proposed method effective. We are encouraged by your remarks on our evaluations. We share your concern regarding the overfitting and fewer-moving-object issue that makes KITTI a less effective benchmark compared to Waymo and nuScenes for unsupervised monocular depth estimation in dynamical scenes. In the following sections, we list our responses to your questions.
>
> ---
> ### Visual Odometry Results
> Table 9 (Rebuttal PDF) compares Monodepthv2, LiteMono, and our proposed methods on their visual odometry performance on both Waymo Open and nuScenes Dataset. For consistency, camera poses from the first 100 test sequences for both datasets as used for evaluation. As shown, our approach achieves comparable results and we will incorporate these results in the final version.
>
> ---
> ### Additional Reference
> Thanks for pointing out [a], as it is relevant to our research direction. Similar to works [4, 9, 15] where semantic knowledge is injected via pretrained segmentation networks to mitigate the negative influence of dynamical objects, [a] utilizes a depth prior injected via a pretrained depth network to do the same. The pretrained depth network in [a] is trained on a large dataset with ground truth depth values, while in contrast, our approach uses no ground truth depth at all. Although KITTI is a suboptimal benchmark for evaluating monocular depth estimation for dynamical scenes, we report the performance of [a] on KITTI, for a consistent comparison with the rest of the mentioned references. We will also discuss [a] in the related works.

---

> > ### Comment · Reviewer_EvaZ · 2023-08-14
> >
> > Thanks for the rebuttal. I read the paper again and I still think that the paper is very great!
> >
> > For the visual odometry, I suggest authors test it in a long sequence, instead of running on 5-frames. For example, you can follow the KITTI odometry benchmark or the TUM RGBD-SLAM benchmark. The experiments are not required at this moment, but adding them to the supplementary can make the paper stronger.

---

> > > ### Author Response · Authors · 2023-08-14
> > > **Response to EvaZ**
> > >
> > > Thank you for your response and support!
> > >
> > > Thanks for the suggestion regarding the visual odometry experiment, and we will include this in the supplementary.

---

### Official Review · Reviewer_9riS · 2023-07-06

**Soundness:** 4 excellent
**Presentation:** 3 good
**Contribution:** 3 good
**Rating:** 6
**Confidence:** 5

**Summary:**

This paper presents a method for monocular depth estimation in dynamic scenes, employing an unsupervised learning approach. While prior work utilizes networks for depth estimation, pose estimation, flow estimation, and mask estimation to decompose object motion and agent motion, the proposed method in this paper similarly adopts these network structures. Distinguishing itself from earlier works, this paper introduces an initial estimation of motion segmentation for the joint learning of depth and independent motion. The main contribution lies in providing an effective initial estimation of motion segmentation, which is critical for determining which pixels in a scene are moving independently. An ablation study demonstrates the effectiveness of the proposed initialization technique. Furthermore, the method achieves state-of-the-art performance on the Waymo and nuScenes datasets.

**Strengths:**

- The proposed method significantly outperforms the state-of-the-art in the Waymo and nuScenes datasets.
- The inclusion of an ablation study effectively illustrates the significance of the proposed initialization techniques.
- The paper introduces a simple yet effective technique that enhances the performance of depth estimation.
- Comprehensive training details are provided, along with evaluations concerning various regions, including static backgrounds, static movable objects, and moving objects.

**Weaknesses:**

- The proposed method fails to surpass the performance of existing approaches on the KITTI dataset.
- The paper primarily focuses on the initialization technique for motion segmentation, which appears overly simplistic and may limit the scope of the paper.

**Questions:**

- Is the proposed network trained separately for each dataset (KITTI, Waymo, nuScenes), resulting in distinct sets of parameters as displayed in Table 1? Alternatively, is there a single trained network that is tested across all three datasets?
- The 'Model and Training setup' subsection appears to describe a training schedule. Is this training schedule uniformly applied across all datasets (KITTI, Waymo, nuScenes), or is it adapted according to specific dataset properties such as size and resolution?

**Limitations:**

Authors adequately addressed the limitations.

---

> ### Author Rebuttal · Authors · 2023-08-10
>
> Thank you for your thoughtful review and for finding the proposed method effective and our evaluation convincing. We list our response to your questions and concerns below.
>
> ---
> ### Fail to Surpass Existing SOTA on KITTI
> Thank you for pointing this out. Although our approach performs only comparably to existing SOTA on KITTI, the lack of moving objects in the KITTI dataset makes it a suboptimal benchmark for evaluating monocular depth estimation for dynamical scenes. In fact, on KITTI, LiteMono [29], without modeling any independent motion, is able to outperform Li et. al [17] and RM-Depth [11], which both explicitly model independent motion. We also offer further insights into this in our manuscript (L237-L239) and supplementary materials (Figure 6). This view is corroborated by *Reviewer EvaZ*, who pointed out that “KITTI dataset has been overfitted, and it contains fewer moving objects than the other two datasets.”
>
> ---
> ### Limited Scope
> The initialization strategy of motion segmentation does not just impact the motion segmentation, but in fact is crucial for all parts of the pipeline. As depicted in Section 4.4, the joint learning of depth, ego-motion, and independent motion is inherently underdetermined, as a family of solutions of depth and object motion would produce accurate reconstruction. Therefore, the proposed initialization of motion segmentation serves as a gradient router that assigns rigid pixels to be modeled by depth and ego-motion via $F_R$ and assigns non-rigid pixels to be modeled by the independent flow field via $F_C$ (Eq. 15). Thus, without using any labels, rigid motion and independent motion can be disambiguated and jointly learned without over-explaining and overfitting to their counterparts when modeling dynamical scenes.
>
> More broadly, while the idea of improving initialization may appear simplistic, this is not the case for underconstrained problems with whole families of solutions. For such problems, we believe that good initialization strategies are not just an engineering trick of limited scope, but are in fact a fundamental component of the solution and deserve careful investigation.
>
> Furthermore, by focusing our attention on the initialization, we are able to create a “simple, yet effective” approach as you point out. We are encouraged that you agree with us that simplicity is a virtue. We could have added more complexity into the architecture or the training paradigm, but we believe that this would have obscured the fundamental insight.
>
> We appreciate your opinion and we will further discuss this point in the introduction and Section 4.4.
>
> ---
> ### Experimental Setup
> The models are trained individually per each dataset, with identical hyper-parameters and the same training paradigm applied uniformly (with exception of input image dimension detailed in L232-L236). The hyperparameters and training paradigm are found on Waymo and directly applied to KITTI and nuScenes without further tuning. We believe that the motion initialization improves training stability and allows generalization across different data distributions with a uniform setup. Further clarification and discussion will be added in the experiment section.

---

> > ### Comment · Reviewer_9riS · 2023-08-17
> > **Thank you for the rebuttal**
> >
> > After carefully reading the rebuttal, I appreciate the clarifications provided. As a result, I have decided to adjust my initial rating upwards. While the main idea of this paper is straightforward, its effectiveness across various works is commendable. Furthermore, the authors adequately address the challenges faced in surpassing the existing state-of-the-art results on KITTI.

---

### Official Review · Reviewer_SBjP · 2023-07-08

**Soundness:** 3 good
**Presentation:** 2 fair
**Contribution:** 2 fair
**Rating:** 4
**Confidence:** 5

**Summary:**

This paper deals with monocular depth estimation by optical flow estimation.
The key is to decompose the flow as ego-motion-induced flow and independent flow.
To implement the idea, depth is predicted on the target image, and camera pose change from target to source is also estimated.
Then a flow network predicts the flow from the target to the source, and a motion mask is also predicted.
Then the final flow that warps the source to the target is recomposed by an alpha combination of the independent flow from the flow network and the rigid flow computed from the depth and the pose change.
Experiments are performed on Kitti, nuScenes, and Waymo Open.

**Strengths:**

Explicit modeling of the ego-motion of the ego car and the independent motion of the moving car can help improve depth estimation.
Since the disentanglement can help optimize the right part of the parameters, otherwise, if there is no explicit modeling, then the network can tradeoff the accuracy on the depth to get an accurate optical flow due to ambiguities.

**Weaknesses:**

The idea of modeling independent motion and ego motion for depth estimation, flow estimation, and moving object estimation is not quite new. For example, this idea has appeared in the following papers:
- "UnOS: Unified Unsupervised Optical-flow and Stereo-depth Estimation by Watching Videos" CVPR 2019.
- "GeoNet: Unsupervised Learning of Dense Depth, Optical Flow and Camera Pose" CVPR 2018
- "Towards scale-aware, robust, and generalizable unsupervised monocular depth estimation by integrating IMU motion dynamics" ECCV 2022
- "Dyna-DM: Dynamic Object-aware Self-supervised Monocular Depth Maps," 2023 IEEE International Conference on Autonomous Robot Systems and Competitions (ICARSC)
- "Learning Optical Flow, Depth, and Scene Flow Without Real-World Labels" ICRA 2023
- "Instance-Aware Multi-Object Self-Supervision for Monocular Depth Prediction" RAL 2022, seems to have better results on Kitti.

**Questions:**

1. "Prima facie, the epipolar ambiguity should not be that much of a problem for a learning-based depth estimation system. This is because independent object motion should be generally inconsistent between different videos, and the depth estimator network cannot recover this information from a single frame alone. Thus, in a hypothetical world filled by moving spheres, the motion information is unobtainable from a single frame and the depth estimation network would indeed regress to the correct depth." can you explain a bit what is the underlying reasoning that the ambiguity should not be a problem? The writing is not explicit, and I have to guess the meaning of words, e.g., "the depth estimator network cannot recover this information" what does "this information" mean? why if the motion is unbotainable then the depth estimation network would indeed regress to the correct depth?
2. please clarify the major contribution of this work.

**Limitations:**

The authors claimed no societal impacts.

---

> ### Author Rebuttal · Authors · 2023-08-10
>
> Thank you for your constructive review and for agreeing with the motivation of our work. We list our response to your questions and concerns below.
>
> ---
> ### Modeling independent motion and ego-motion is not new
> Compared to the mentioned papers, our method learns from **monocular** videos only **without any labels or semantic knowledge** from pretrained models and jointly predicts monocular depth, ego-motion, 3D independent flow, and motion segmentation to disambiguate between the rigid and non-rigid parts of a dynamical scene. The following bullet points detail the differences in the methodology and the table of results are provided in Table 8 (Rebuttal PDF).
>
> - UnOS (Wang et al.) uses **stereo input** for both training and inference, which is a different task from our method which learns from monocular videos only.
>
> - DRAFT (Guizilini et al.) utilizes **additional synthetic data with synthetic ground truth**, whereas our method only learns from the unlabeled monocular videos. Additionally, **multiple frames** are used to predict monocular depth for DRAFT, where our approach only uses a single frame.
>
> - Zhang et al. uses **additional inertial measurement unit (IMU) measurements** for training. In comparison, our method only has access to monocular videos and explicitly models independent motion field and motion segmentation to aid depth learning. We achieve comparable results on KITTI without accessing additional supervisions.
>
> - Dyna-DM (Saunders et al.) uses a **background mask from an off-the-shelf pretrained Mask R-CNN** to reason with dynamical objects. In comparison, our method does not use any semantic knowledge from pretrained networks and reason about dynamical objects solely by disentangling the underlying motion. Quantitatively, our approach consistently outperforms Dyna-DM on KITTI.
>
> - Boulahbal et al. [2] uses a **frozen pretrained segmentation network trained on Cityscape** to predict piecewise rigid pose for each dynamic object, whereas our method does not leverage any semantic knowledge from pretrained networks. Our method is comparable with [2], despite not obtaining object information from pretrained networks.
>
> - GeoNet [28] (Yin et al.) only **models moving objects in 2D** via optical flow, in contrast, we model the behavior of moving objects via a 3D motion field. Modeling the 3D motion allows our model to accurately learn the depth of independently moving objects by explaining its overall flow as a combination of independent motion and ego-motion, rather than ignoring the dynamical pixels in the 2D case. In addition, the consistency mask predicted by [28] also includes occlusions and non-Lambertian surfaces, which is different from motion segmentation in our case. Quantitatively, our method significantly outperforms [28] on KITTI.
>
> Thank you for pointing out the references, the appropriate discussion and comparison will be incorporated in the final draft.
>
> ---
> ### Clarify Contribution
> We list the following four main contributions for our work:
> - Our method learns from **monocular** videos only **without any labels or semantic knowledge** from pretrained models and jointly predicts monocular depth, ego-motion, 3D independent flow, and motion segmentation.
>
> - We reformulate the independent 3D motion field as the difference between the rigid flow induced by ego-motion and complete scene flow to enable proper regularization and facilitate learning.
>
> - We propose a **novel motion initialization** technique that resolves the ambiguity between camera motion and independent motion that frequently appears in complex dynamical scenes. This initialization allows the proposed architecture to be jointly trained without any labels.
>
> - Our proposed method improves unsupervised monocular depth performance for dynamical scenes, especially for moving objects, as benchmarked on both Waymo Open and nuScenes.
>
> ---
> ### Clarify “Prima facie…” on L115-L120
> Here is a rephrasing of that paragraph:
>
> The depth network gets a single frame as input and must predict where objects are in the 3D scene. The training objective is that when the camera moves, the next frame must be correctly reconstructed.
>
> For independently moving objects, even in the ideal scenario when the depth work predicts the right depth for these dynamical objects, their predicted position in the next frame will still be incorrect due to their own motion. However, the same object will move one way in one video, another way in another video and be static in a third. Since the moving instances are unpredictable and cannot be modeled by the depth network from a single frame, the reconstruction error caused by their independent motion cannot be avoided.
>
> Therefore, if there is no a priori bias in the direction or speed of object movement in the data, then the best that the depth network can do is to fit the static instances well and thus learn to predict the correct depth. This is what we mean by the depth network not being able to recover “this information” which is the object's motion.
>
> Our point is that unfortunately, in self-driving scenes, there are statistical regularities in object motion. For example, a car viewed from the back will mostly be moving away from the camera. This means that the depth network has the ability to predict this movement from a single frame and alter its depth prediction (by placing the car farther away than the true depth) as shown in Figure 1(a) to reconstruct the next frame correctly. Conversely, when one sees the frontal view of a car, it is likely that the car is moving towards the camera and the depth network can predict the car to be closer to explain the pixel motion, as shown in Figure 1(b). Additionally, the speed of the object can be roughly estimated from object identity (pedestrian < cyclist < car) and background context (residential < city < highway).
>
> We hope this helps and will update the phrasing.

---

> > ### Comment · Reviewer_SBjP · 2023-08-14
> >
> > Thanks for the response. Still not clear of the contribution of the method. In the pipeline, pre-trained networks are also used as in other works, please elaborate more on the exact information that the framework leverages.

---

> > > ### Comment · Reviewer_EvaZ · 2023-08-14
> > > **Response to SBjP**
> > >
> > > Please note that the authors emphasize that they do not use a pre-trained network in the propose pipeline.

---

> > > ### Author Response · Authors · 2023-08-14
> > > **Response to SBjP**
> > >
> > > Thank you for your response!
> > >
> > > As *Reviewer EvaZ* pointed out, our approach does not use any auxiliary pretrained models during training. This is in contrast with [2], [4], [9], [15], SC-DepthV3, and Dyna-DM, where the auxiliary models used during training are pretrained directly or approximately for the task in question.
> > >
> > > Specifically, [2], [4], [9], [15], and Dyna-DM use segmentation masks output by a frozen pretrained segmentation network in their pipelines and loss functions during training, while SC-DepthV3 uses the predictions of a pre-trained depth network to enforce a depth prior during training. In contrast, our main result is that by only training on unlabeled videos, we can disambiguate object motion and estimate correct monocular depth for dynamic scenes, without any help on obtaining object or depth information.
> > >
> > > Nevertheless, we have provided the comparisons on KITTI in the Rebuttal PDF. In Table 8, our proposed method consistently outperforms against Dyna-DM, while being comparable to [2], despite not obtaining any semantic information from segmentation networks.
> > >
> > > As a clarification, for most methods compared, the depth encoder starts training with ImageNet weights. We do the same to be commensurate with Monodepth2[8] and LiteMono[29]. We could also initialize the weights via a self-supervised objective, but for consistency with existing works, we use ImageNet weights at initialization.
> > >
> > > The comparisons and clarifications will be added to the camera-ready version and we promise to incorporate our model performance when it is trained from scratch without ImageNet weights with appropriate baselines.

---

### Author Rebuttal · Authors · 2023-08-09

We thank the reviewers for their insightful comments and positive feedback. We are encouraged that they found our work well-written (*Reviewer EvaZ, 9pRa*), well-motivated (*Reviewer SBjP*), effective (*Reviewer 9riS, EvaZ, 9pRa*), and with proper evaluations (*Reviewer 9riS, EvaZ*).

We appreciate all of the paper references mentioned in the reviews. To be clear, in our work, we aim to solve **unsupervised monocular depth estimation in dynamical scenes**. At training time, we only have access to a set of monocular videos and assume no labels of any kind (synthetic or real), no stereo information, no other input modalities, and no auxiliary pretrained models. At inference time, monocular depth is predicted from a single frame without any stereo or sequences of frames.

We are encouraged that the mentioned works address dynamical scenes in some capacities, however, most are tackling different problem setups that make it easier to disentangle and mitigate the presence of dynamical objects.

- ***stereo-view during training***. Stereo-view gives significant information to depth without needing to learn from camera motion during training. Once depth can be estimated, the independent motion that is entangled with rigid-motion, especially collinear with camera motion, can be disambiguated as well. In contrast, in our work, we are able to learn depth jointly with object motion from monocular videos only without using any labels.

- ***additional modalities during training***. More broadly, access to additional modalities can reduce the learning complexity by adding additional information to the observed videos. For instance, camera motion can be found via an inertial measurement unit (IMU), without being estimated from the apparent motion in the videos. In our setting, the apparent motion cues are the sole input and supervising signal for learning.

- ***auxiliary models during training***. Auxiliary models pretrained on instance segmentation or monocular depth estimation in a supervised manner can provide additional information for learning to model dynamical scenes, such as providing the region containing moveable objects or a depth prior for dynamical objects. In comparison, our method learns depth, independent motion, and motion segmentation solely from observing monocular videos without any additional pretrained models.

- ***labels during training***. With access to ground truth labels, the learning problem can, to some extent, rely on the supervised objective, which is much easier than our learning objective that is purely based on image reconstruction. In cases of ground truth depth labels (real or synthetic), similar to stereo-view during training, independent motion can be directly disambiguated given the known depth information.

- ***stereo-view during inference***. Scene geometry and depth information can be extracted via simple correspondence, which is a much easier setup compared to monocular input during inference.

- ***multi-frame during inference***. Additional consecutive frames would introduce much more information regarding scene geometry that is not retrievable from a single frame as input during inference.

Nevertheless, we provide the results for comparison in Table 8 (Rebuttal PDF) and we discuss the mentioned works further in the individual responses.
Since previous works have been primarily focused and reported on KITTI, we report comparisons on KITTI, even though KITTI has fewer moving objects compared to Waymo and nuScenes (pointed out by *Reviewer EvaZ*, L237-L239 in manuscript and Figure 6 in supplementary materials). For additional qualitative results in dynamical scenes, please refer to our submitted supplementary materials containing 12 video visualizations of our model’s predictions.

We look forward to your response and we are happy to address any additional questions or concerns!

---

### Decision · Program_Chairs · 2023-09-21

**Decision:**

Accept (poster)

**Comment:**

The paper received conflicting reviews, which could not be reconciled during the post-rebuttal discussion. Two (experienced) reviewers strongly support the paper, and a third one is also positive. On the contrary, two reviewers argue against the paper. The AC has read the paper and checked the discussion, and found that some of their negative arguments are not valid. Most importantly, the correct observation that "modelling independent motion and ego motion for depth estimation, flow estimation, and moving object estimation is not quite new" does not imply a lack of novelty: that the high-level ideas have appeared in other works could be said about the overwhelming majority of NeurIPS publications. Also the fact that foundation models are available does not imply that one must use them and that starting from "only" ImageNet pre-training is wrong. The AC concluded that while the contribution of the paper is not ground-breaking, it is sufficiently interesting to warrant publication.